



# A critical review and presentation of the complete, historic series of K-indices as determined at Norwegian Magnetic Observatories since 1939

Ingeborg Frøystein[1,2] and Magnar Gullikstad Johnsen[2]

[1]Department of Physics and Technology, UiT the Arctic University of Norway, Tromsø, Norway
[2]Tromsø Geophysical Observatory, Faculty of Science and Technology, UiT The Arctic University of Norway, Tromsø, Norway

**Correspondence:** M.G. Johnsen (Magnar.G.Johnsen@uit.no)

**Abstract.** The complete, existing, time series of K-indices from Norwegian observatories in Tromsø (TRO), Dombås (DOB) and Bear Island (BJN) has been digitized. The digitized time series are continuous spanning from 1939 (DOB) and 1947 (TRO) until 1998. Today, Tromsø Geophysical Observatory manages geomagnetic observations throughout Norway and K-indices are calculated in real-time with a fully automatic, in-house method. In t his paper, the old, hand-scaled, and new, automatic, time series of K-indices are reviewed and compared for the intervals were they overlap. Our analysis confirms that the digital K-index series is a valid continuation of the old series. Since 1939, three K-index derivation methods have been applied to Norwegian magnetic observatory data. These are traditional hand-scaling, the method developed by the Finnish Meteorological Institute and an in-house method. Here, we compare the tree methods. It becomes clear that each method both have strengths and weaknesses. Importantly, differences arise when calculating the quiet-day variation, especially during periods of consecutive disturbed nights at auroral latitudes. By analysis of the K-index frequency distributions for six stations in mainland Norway and on Svalbard, it arises that the lower limit for $K = 9$ of 2000 nT is too high for TRO and $K = 9$ of 750 nT possibly too low at DOB. The assumption that X > Y, which makes it possible to calculate $K$ for only the magnetic X component is investigated, and it is shown that the assumption is indeed only valid for auroral stations. In total, this paper presents all K-indices derived from Norwegian observatories since the nineteen-thirties until today, the used derivation methods and the long, historic time-series as a whole, and thus, enables a critical use of the indices for future scientific work.

## 1 Introduction

Today, Tromsø Geophysical Observatory (TGO) at UiT the Arctic University of Norway takes care of geomagnetic observations throughout Norway, spanning sub auroral, auroral and polar cap latitudes. The first systematic geomagnetic measurements in Norway started with Kristoffer Hansteen and the establishment of the Kristiania Magnetic Observatory in 1843 (Wasser-



fall, 1941). The time series from this observatory does not include continuous registrations of the magnetic elements, but rather two absolute measurements every day. Kristian Birkeland introduced variometers in geomagnetic work in Norway. Such instruments were installed during both his expeditions to Northern Norway in 1899-1900 (Birkeland, 1901) and 1902-1903 (Birkeland, 1908).

From 1912 the Haldde Observatory in Alta was operated as a permanent "magnetic-meteorological" observatory under the leadership of Ole Andreas Krogness, one of Birkeland's former students. The Haldde observatory performed continuous magnetic registrations until it was closed down in 1926 (Krogness and Tönsberg, 1936). The Haldde time series was continued at the Auroral Observatory in Tromsø (TRO) from 1928, after a temporary period at the Geophysical Institute nearby. For acquiring context to the work at Haldde from a sub-auroral station, the Dombås Magnetic Observatory (DOB) was established
in 1916. This is today the oldest magnetic observatory in operation in Norway.

During the International Polar Year (IPY) 1932-33, Polish researchers set up a magnetic observatory on Bear Island (BJN). The measurements were continued by the Auroral Observatory in Tromsø with the assistance of the Meteorological Institute. With an intermission during World War II (1941-1945), the measurements still continues today.

Thus, measurements at Norwegian stations DOB, TRO and BJN are closing in on a century of magnetic measurements
stretching from sub-auroral to auroral latitudes. Prior to the digital era, efficient ways of providing measurements were needed, to allow workers to study and combine data from multiple locations away from their own. The efficient way to do this was to compile indices from the raw magnetic measurements and disseminate them via yearbooks and exchange networks of these. A wide range of indices have been used for various purposes and analysis aims. Still, today, even with full access to sub-minute resolution geomagnetic data over the internet, indices have maintained their popularity. Even if the true data usually give a more
accurate and physical interpretable picture of the geomagnetic activity, and studies such as e.g. Menvielle et al. (2011) warns about the validity at high latitudes, owing to considerations regarding the magnetic energy density of the magnetic variations, the K-index in particular is widely used.

Bartels et al. (1939) introduced the K-index and through his and co-workers' efforts it became the de-facto standard measure for local geomagnetic disturbances among magnetic observatories worldwide. In this paper we present efforts made to digitize
time series of geomagnetic activity (K-index) at three magnetic observatories in Norway as far back in time as possible. We document the sources of where these data have been found, and based on these, present the time series including a time series analysis. We analyse the transition from manual to automatic scaling of the K-index to ensure that the time series can be assumed to be continuous. Furthermore, we investigate the difference between the IAGA endorsed FMI method for K-index scaling and an in-house method, termed the TGO method. We find that, at least for the latitudes in question here, the TGO
method does not perform worse than the FMI method.

Through the work presented in this paper, we make the complete, historic time series of geomagnetic activity as described using the K-index, available to the scientific community. By discussing and analysing the methods that have been used and the transition to the digital area, we pave the way for critical, scientific application of the time series.





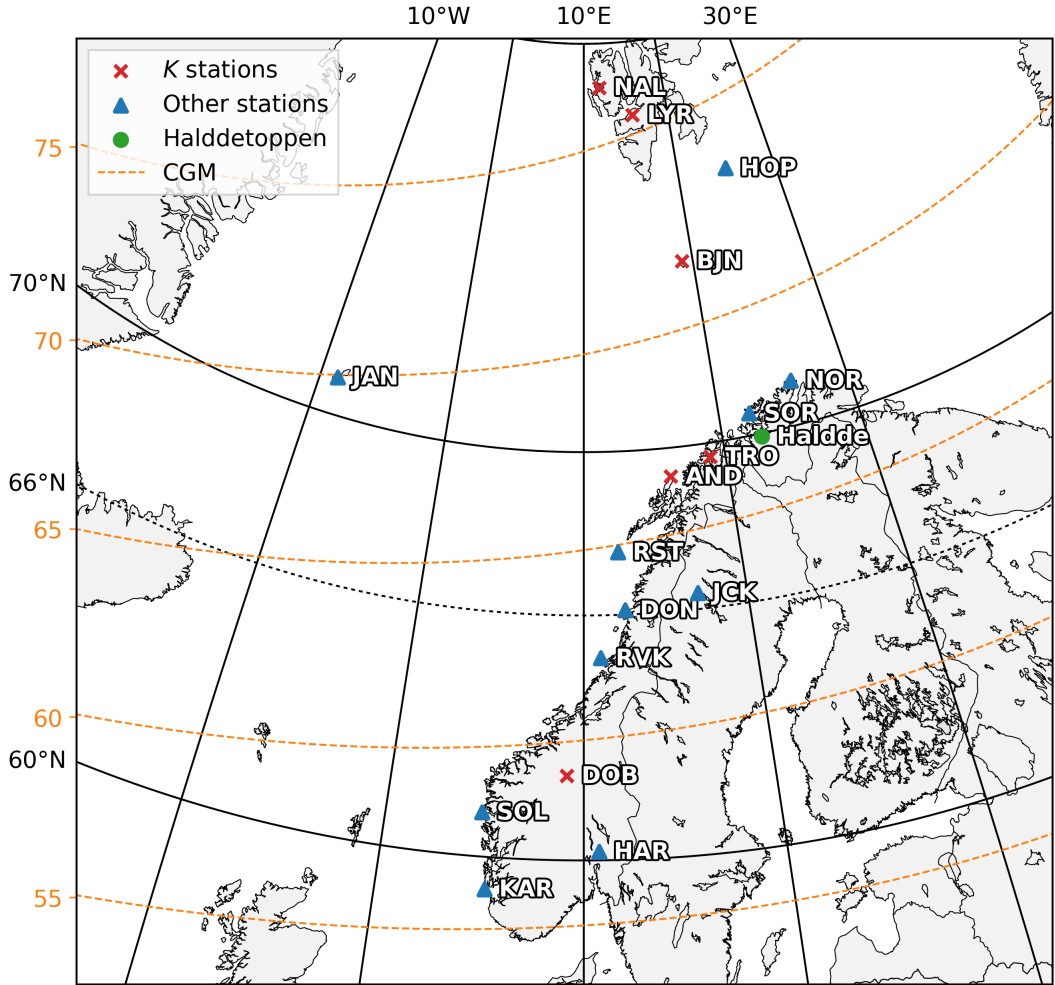

**Figure 1.** Map of Norwegian magnetic stations. $K$-index stations are marked with red crosses. Remaining stations are market with blue triangles. The historic observatory at Halddetoppen is marked with a green circle. CGM latitudes in orange dashed lines. See Table 1 for details.

## 1.1 Magnetometer stations in Norway

Currently, TGO operates a total of 17 variometer stations and observatories, from Karmøy in the south-western part of Norway to Ny-Ålesund on Svalbard. All 17 stations are listed in Table 1, along with their coordinates, the station type and the first year(s) of operation. A map of the station locations is shown in Figure 1. The map also indicates the location of the historic observatory at Halddetoppen, marked in white.





**Table 1.** Information on the geomagnetic observatories and stations in Norway. Corrected geomagnetic coordinates and L-values from the 2022 IGRF/DGRF model (NASA Goddard Space Flight Center, Corrected Geomagnetic Coordinates and IGRF/DGRF Model Parameters, https://omniweb.gsfc.nasa.gov/vitmo/cgm.html))

| Code | Name | GCS coord. | | CGM coord. | | L | Type | Start year |
|------|------|-----------|---------|-----------|---------|------|------|------------|
| NAL | Ny-Ålesund | 78.92 N | 11.93 E | 76.84 | 106.59 | N/A | Calibrated variometer | 1966 |
| LYR | Longyearbyen | 78.20 N | 15.83 E | 75.92 | 107.96 | N/A | Calibrated variometer | 1993 |
| HOP | Hopen | 76.51 N | 25.01 E | 73.81 | 112.06 | 13.06 | Calibrated variometer | 1988 |
| BJN | Bear Island | 74.50 N | 19.00 E | 72.04 | 104.82 | 10.68 | Observatory | 1932/1948 |
| NOR | Nordkapp | 71.09 N | 25.79 E | 68.34 | 107.17 | 7.45 | Calibrated variometer | 2007 |
| JAN | Jan Mayen | 70.90 N | 8.7 W | 70.16 | 79.57 | 8.82 | Variometer | 1928-1936/1988 |
| SOR | Sørøya | 70.54 N | 22.22 E | 67.90 | 103.85 | 7.17 | Calibrated variometer | 2001 |
| TRO | Tromsø | 69.66 N | 18.94 E | 67.11 | 100.55 | 6.71 | Observatory | 1928 |
| AND | Andenes | 69.30 N | 16.03 E | 66.86 | 97.93 | 6.57 | Calibrated variometer | 1996 |
| RST | Røst | 67.53 N | 12.10 E | 65.16 | 93.55 | 5.75 | Calibrated variometer | 2017 |
| JCK | Jäckvik (Sweden) | 66.40 N | 16.98 E | 63.77 | 96.90 | 5.20 | Calibrated Variometer | 2010 |
| DON | Dønna | 66.11 N | 12.50 E | 63.63 | 93.06 | 5.15 | Calibrated variometer | 2007 |
| RVK | Rørvik | 64.95 N | 10.99 E | 62.46 | 91.21 | 4.75 | Calibrated variometer | 1998 |
| DOB | Dombås | 62.07 N | 9.11 E | 59.41 | 88.31 | 3.92 | Observatory | 1916 |
| SOL | Solund | 61.08 N | 4.84 E | 58.50 | 84.37 | 3.72 | Calibrated variometer | 2004 |
| HAR | Harestua | 60.21 N | 10.75 E | 57.30 | 88.92 | 3.48 | Calibrated variometer | 2017 |
| KAR | Karmøy | 59.21 N | 5.24 E | 56.38 | 83.95 | 3.31 | Calibrated variometer | 2003 |

### 1.1.1 The $K$ index

The $K$ index (Bartels et al., 1939) is a station-specific measure of the geomagnetic disturbance, or rather disturbance range, over a 3 hour interval. The $K$ index only includes the geomagnetic disturbance and not quiet day variations (Matzka et al., 2021). It ranges from $K = 0$ to $K = 9$ on a quasi-logarithmic scale. The value is calculated for intervals 00-03, 03-06, ..., 21-24 UT. Previously, all components D, H and Z were used when calculating $K$. Today, only D and H is used, partly to remove the effect of induced underground currents (Mayaud and IAGA, 1967). Before a complete set of directions for calculating $K$

was published by Mayaud and IAGA (1967), different methods for hand scaling $K$ had been used at different observatories. A rigorous description of the method for hand-scaling is given by Mayaud and IAGA (1967), but in short the steps can be described as follows.

The first step is to estimate the quiet day variation $S_q$ or Quiet Day Curve (QDC). When the QDC is found, the $K$ values for components H and D, or X and Y, on each 3h interval is found directly by using a customized gauge. The gauge is made to fit

the ranges for each $K$ value for the observatory. See e.g. Figure 5 in Mayaud and IAGA (1967) for such a gauge. The resulting $K$ indices, and especially the estimated QDC, will vary somewhat based on the observers, their experience and knowledge of




**Table 2.** Lower limits (nT) for each $K$ value for the Niemegk observatory (Mayaud, 1980) and frequency distribution (Matzka et al., 2021).

| Lower limit (nT) | 0 | 5 | 20 | 20 | 40 | 70 | 120 | 200 | 330 | 500 |
|---|---|---|---|---|---|---|---|---|---|---|
| $K$ value | 0 | 1 | 2 | 3 | 4 | 5 | 6 | 7 | 8 | 9 |
| Frequency, [%] | 7.116 | 24.42 | 29.42 | 23.15 | 10.62 | 4.06 | 0.966 | 0.188 | 0.0418 | 0.019 |

**Table 3.** Conversion table for $K$ to the ak index.

| $K$ | 0 | 1 | 2 | 3 | 4 | 5 | 6 | 7 | 8 | 9 |
|---|---|---|---|---|---|---|---|---|---|---|
| Ak value | 0 | 3 | 7 | 15 | 27 | 48 | 80 | 140 | 240 | 400 |

the quiet day variation at their observatory. However, Matzka et al. (2021) notes that it is expected that the agreement between $K$ indices derived by different experienced observers generally would be at least 80 %.

Depending on the geomagnetic latitude and therefore the amount of geomagnetic disturbance experienced at each station, the ranges for each station are adjusted to the latitude. This is done by scaling the Niemegk-ranges, shown in Table 2, by a set lower limit for $K = 9$, the K9-limit.

The $ak$ index is derived from $K$, converting $K$ to a linear scale (Bartels and Veldkamp, 1954; Van Sabben and IAGA, 1972). The conversion values are shown in Table 3. It is therefore possible to compute averages: daily $Ak$, monthly $Ak$ and yearly $Ak$. These indices are useful when investigating long-term variation of geomagnetic disturbances (e.g. (Nevanlinna, 2004; Nevanlinna et al., 2011)))

The $K$ index is routinely calculated for 6 stations in Norway. These 6 stations are, sorted from south to north, Dombås (DOB), Andenes (AND), Tromsø (TRO), Bear Island (BJN), Longyearbyen (LYR) and Ny-Ålesund (NAL). The stations are marked in red on the map in Fig. 1. $K$ indices were first published in Norway in 1939, from the DOB observatory. From 1947 $K$ indices were also published from the TRO observatory. $K$ indices from TRO and DOB are calculated to this day. $K$ indices were calculated for BJN during a short interval from 1951 to 1965 and also for the International Polar Year (1932-33). In more recent years, namely in 1986, 1993 and 1996, calculation of the the $K$ index was started at LYR, NAL and AND, respectively. The calculations of $K$ indices at LYR and NAL were started based on popular request.

Due to the large spread in geomagnetic latitude among the Norwegian $K$ index stations, 3 different $K$9-limits, spanning from 750 nT to 2000 nT are used. The limits are presented in Table 4.

Norwegian $K$ indices have been used in numerous studies, especially since the TGO method was implemented and the $K$ indices were made available digitally. The studies include, to name a few, work on the aurora (Nanjo et al., 2022), work on Polar Mesospheric Summer Echoes (Bremer et al., 2000, 2001; Zeller and Bremer, 2009), GNSS disturbances and scintillation (Andalsvik and Jacobsen, 2014), thermal structures in the ionosphere, thermosphere and mesosphere (Kurihara et al., 2010) and dynamic instabilities and their relationship to geomagnetic activity (Nozawa et al., 2023).



**Table 4.** $K$9-limts for the Norwegian $K$ index stations.

| Code | Name | $K$9 Limit (nT) |
|------|------|-----------------|
| NAL | Ny-Ålesund | 1800 |
| LYR | Longyearbyen | 1800 |
| BJN | Bear Island | 2000 |
| TRO | Tromsø | 2000 |
| AND | Andenes | 2000 |
| DOB | Dombås | 750 |

## 2 The data set, digitization and $K$ derivation methods

### 2.1 $K$ indices

The $K$ index from the observatories in Tromsø, Dombås and Bear Island has been published in (1) the IAGA Bulletin no. 12 (available at International Service of Geomagnetic Indices, IAGA at http://isgi.unistra.fr/iaga_bulletin.php) and in (2) year-books from the observatories in Dombås and Tromsø. All yearbooks have been scanned and published on the TGO website (https://www.tgo.uit.no/ScanRap/GeoPhysRap.html). $K$ indices from Tromsø were submitted to IAGA from 1947 and $K$ indices from Dombås were submitted from 1946. $K$ indices from Bear Island were submitted only on a short interval from late 1957 to mid 1959, as a contribution to the International Geophysical Year (IGY). Indices from the International Polar year (1932-33) were also submitted to IAGA for all three stations.

The yearbooks from Tromsø were published by the Norwegian Institute of Cosmic Physics from 1930 to 1965 and by the Auroral Observatory, the University of Tromsø from 1966 to 1998. The Observations from Bear Island are published in the Tromsø yearbooks. The yearbooks from Dombås were published by the Norwegian Institute of Cosmic Physics from 1916 to 1958, by the Geophysical Institute at the University of Bergen, Norway from 1959 to 1988 and the Institute of Solid Earth Physics and Geomagnetism at University of Bergen, Norway from 1989 to 1998. No yearbook was made for DOB on the interval 1949-1951. In this period the observatory was in the process of being moved (Gjellestad et al., 1957).

The $K$ indices from all three observatories were digitized by typing the published values into plain ASCII-files. When digitizing data manually, by "human hand", it is impossible to completely safeguard against errors such as typing the wrong number. However, for this data set it was the most convenient way owing to the large number of different table formats and page layouts in both the yearbooks and the IAGA Bulletin no. 12. Table B1 show were the $K$ values were published. The digitized $K$ values were retrieved from the observatory yearbooks when $K$ indices were included in the yearbook for that station and year. Where $K$ indices were not included in the yearbook, the $K$ indices were obtained from the IAGA Bulletin no. 12.

The digitization of historic data and dissemination over the internet has gained an increased popularity over the decades (e.g. Sergeyeva et al. (2021); Nevanlinna and Häkkinen (2010)). This is important in order to make valuable, historic, and often high quality, data available to a wider scientific public than only those with direct access to the original material. Therefore



the digitized, Norwegian $K$ indices have been made public through the TGO web pages next to the more recently generated
geomagnetic data (available at https://flux.phys.uit.no/Kindice/).

## 2.2 Digital derivation methods used in Norway

Traditional hand-scaling of magnetograms (Bartels, 1957; Mayaud and IAGA, 1967) was used for deriving $K$ for TRO from
1947-1991, for BJN from 1951-1965 and for DOB from 1939-1994. In addition, two digital derivation methods have been used
to derive $K$ indices in Norway. The two methods are the FMI method and an in-house method from now on referred to as the
TGO method. Brief descriptions of the two methods are presented in the following sections.

### 2.2.1 The FMI method

The FMI method, developed by Lasse Häkkinen at the Finnish Meteorological Institute (Sucksdorff et al., 1991), was in 1993
officially recognized as a method for deriving $K$ by the International Association of Geomagnetism and Aeronomy (IAGA
News, December 1993). The FMI method was used between 1995-1998 for deriving $K$ at DOB (Institute of Solid Earth
Physics, Geomagnetism, 1997). The FMI method FORTRAN code is available from The International Service of Geomagnetic
Indices (ISGI) (The International Service of Geomagnetic Indices (ISGI), $K$ indices softwares, https://isgi.unistra.fr/softwares.
php) and has been applied to digital TGO data in this study.

The FMI method is an iterative scheme based on linear elimination for calculating the $K$ values, where the $K$ values and
QDC for one day is found based on magnetograms from the previous, current and next day. Descriptions of the method
is given in Sucksdorff et al. (1991); Menvielle et al. (1995) and also by detailed examples on the FMI web pages (https:
//space.fmi.fi/MAGN/K-index/FMI_method/). In short, the method can be summarized as follows. First, preliminary $K$ indices
are calculated for all 8 intervals without first subtracting a QDC. These preliminary $K$ indices are then used in calculation of the
first fitted QDC. Next, new $K$ indices are calculated after subtracting the fitted QDC. These $K$ indices are used in calculation
of the final fitted QDC. The final fitted QDC is subtracted and the final set of $K$ indices are calculated. This scheme is done for
both the X and Y components and the largest value of the two in each interval is then selected.

### 2.2.2 The TGO method

The TGO method, an automatic algorithm, was developed by Truls Lynne Hansen at the Auroral Observatory (today; TGO),
as a light-weight and simpler alternative to the FMI method especially aimed at real-time generation (Truls Lynne Hansen,
personal communication). The TGO method is used on all six Norwegian $K$ index stations, it calculates the values in real-time
and therefore in principle applies to any data where digital data exist. This means that K-indices are available by the TGO
method for NAL since 1986, TRO since 1987, AND since 1995, LYR since 1993, BJN since 1987 and DOB since 1993. In
addition, the method was used, in an early form (from now referred to as the TGO' method), on TRO data and published in the
yearbooks from 1992 to 1998. The TGO method is described below.





The TGO method uses the fact that the quiet day variation is regular and therefore should be relatively trivial to predict owing to its strong relation to time of day and time of year. Static Quiet Day Values (QDVs) are used for correcting the magnetograms when calculating $K$ by the TGO method. The QDVs are are the monthly averages of the hourly quiet day values, of the horizontal component (H), covering the entirety of solar cycle 22, resulting in 12x24 correction values. Using the means over an entire solar cycle removes the possible problem of subjectivity when selecting quiet days for use in the estimation of the QDC (Valach et al., 2016); the QDC is simply predetermined. However, the variation in amplitude over the solar cycle, owing to varying solar irradiance, will introduce an uncertainty of about +/- 10 nT. The QDCs for TRO and DOB are shown in Fig. 2. The quiet day values that have been averaged over are published in the yearbooks from 1987 to 1999, from TRO and DOB. TRO and AND is corrected by QDV from TRO because they are only separated by approximately 120 km and 0.3 ° in geomagnetic latitude. DOB is corrected by the QDVs from DOB. BJN, LYR and NAL are uncorrected. At these high latitudes, omitting QDC correction will only introduce a small uncertainty owing to the large disturbances generally experienced from the auroral electrojets, and correspondingly small QDC amplitudes, here.

Because all the Norwegian $K$ index stations are at relatively high latitudes, only the H-component is used when deriving K. This is based on the assumption that it is reasonable to assume that H > D, or X > Y at high latitudes. Every 1-minute resolution data point of the H component is corrected by a smoothed and interpolated value of the QDV. It is fair to be sceptical against this assumption for at least DOB.

After the H-component is corrected (if applicable), the range, $r$, of the variation over each 3h interval is calculated. $r$ is then matched to $K$ following the K-bins resulting from scaling the Niemegk bins (Matzka et al., 2021) to fit the K9-limits at each station, given in Table 4. A perk of the TGO method, compared to hand-scaling and the FMI method, is that K-values can be calculated in near real-time. This is because the method does not rely on the next day of magnetograms (like the FMI method) or even the full day (like HS).

In the TGO method, all data gaps are allowed and the gaps are never interpolated. This works well because the TGO method only uses the H component data in simple operations, that is subtracting the QDC and finding the maximum range $r$ per interval. From a technical viewpoint, missing data is therefore not a problem. However, this will only yield representative $K$ values if the data gaps on an interval are small. If larger gaps occur, it is likely that the TGO method will assign a lower $K$ value. However, larger gaps will never lead to a too large $K$ value. This means that, in the event of abundant datagaps, we should expect that $K$ values derived by the TGO method is slightly smaller than those hand-scaled.

## 3 Frequency distributions of $K$

Frequency distributions for $K$ should be log-normal-esque with a peak frequency at $K =2$, following the standard $K$-index observatory in Niemegk (NGK). Also, values of $K = 2$, 3 and 4 should account for >50 % of cases. See Matzka et al. (2021) for further details on the ideal distribution. An appropriate $K9$-limit will result in well-fitting bins for each $K$-value, meaning the bins are a good fit for representing the K-variation experienced at the station in question. Figure 3 show distributions of





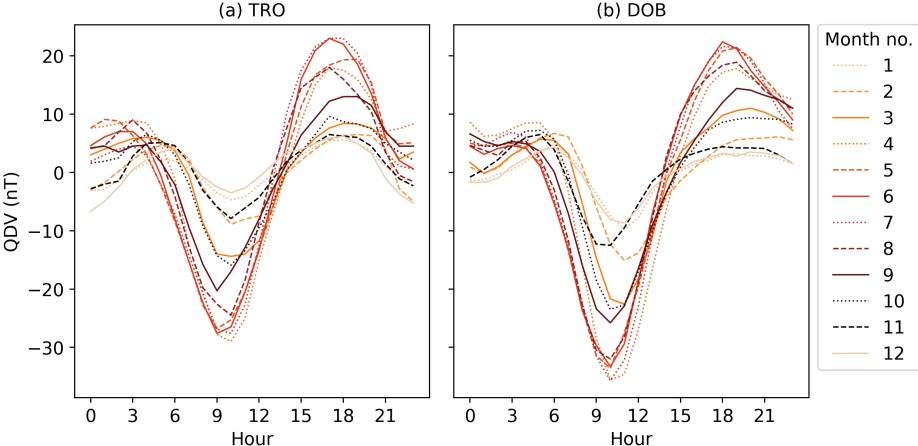

**Figure 2.** Quiet day values (QDV) for a) TRO and b) DOB.

K-values for all six $K$ index stations in Norway, AND, TRO, DOB, BJN, LYR and NAL. All available $K$ indices for each stations are used in the corresponding histograms. The frequency distributions are also presented in Table A1.

There are several points of interest concerning the distributions. We note that the distributions for BJN, LYR and NAL are similar in shape. All three distributions are well tapered in both ends of the range for $K$ and have the maximum frequency at $K = 3$. This is close to the Niemegk-distribution. It is also expected that the distribution for LYR includes slightly lower frequencies for low $K$ values and higher frequencies of high $K$ values than NAL. NAL is further into the Polar Cap than LYR, and it is reasonable to expect that LYR therefore experiences more geomagnetic variation due to the position closer to the auroral oval. This direct comparison is only possible because LYR and NAL share the same $K9$-limit (1800 nT). The DOB distribution is clearly log-normal-esque with a peak frequency at $K = 2$. However, the distribution includes a very large frequency of $K = 9$ which requires further investigation.

The distribution for AND is drastically different from the remaining distributions. The same shape is seen for TRO on the same interval as AND, 1996-2021. The frequencies are descreasing with increasing $K$, the most frequent $K$ value being $K = 0$. Even though the $K = 0$-frequency is large, we still have low, expected frequencies of $K = 9$. The histogram for TRO improves when the whole interval is included. However, the skewed shape for TRO and AND cannot be ignored. The interval 1998-2021 includes only TGO derived $K$ indices. As discussed in section 2.2.2, it is expected that the TGO method will assign lower values for $K$ than HS. It is therefore of interest to, in more detail, inspect the overlapping period of $K$ values for TRO and DOB during the transition between handscaling and the TGO method. Another option is that the shape is influenced by the included years and therefore the solar activity of the period 1947-1996 versus 1996-2021. This option will also be investigated.

In addition the the distribution features shown and discussed here, the distributions also exhibit expected seasonal and diurnal variation. That is, the frequencies of larger $K$ values maximize during spring and autumn, which is consistent with the Russell-





McPherron effect (Russell and McPherron, 1973). The frequencies of larger $K$ values also maximize during the night (21 UT - 6 UT), consistent with substorm activity during the night.

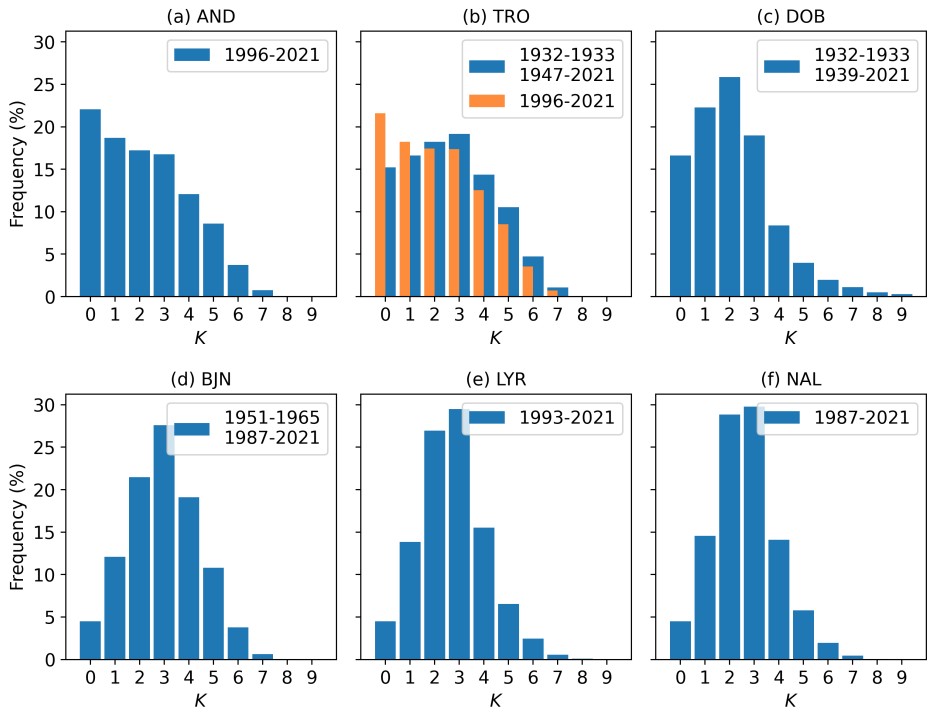

**Figure 3.** Histograms of $K$ values for a) AND, b) TRO, c) DOB, d) BJN, e) LYR and f) NAL. The exact frequencies are shown in Table A1. The interval for each histogram is shown in each legend.

## 3.1 Distributions during the transition from handscaling to automatic methods

Figure 4 and Table A2 show distributions for $K$ for TRO on the overlap interval 1988-1991 and for DOB on the overlap interval 1993-1998. $K$ values derived by HS/FMI are shown in blue, TGO derived $K$ values are shown in orange. For TRO, the shapes of the two distributions are similar. We note, as expected, that the TGO method includes a higher frequency of $K = 0$. However, the distributions show no indication that the skewness for TRO and AND in Fig. 3 is a feature solely of the TGO method.

The general shapes of the distributions are also similar for DOB. However, we note somewhat larger differences between the two distributions. The differences are not systematic, but they are clearly larger for smaller $K$ values (<4). It is not immediately clear why these differences occur, but a possible explanation is that due to the low $K$9-limit of only 750 nT, resulting in small bins of e.g. 0-7.5 nT and 7.5-15 nT, the estimation of QDC is of greater importance than for TRO. Due to the small bins there is a larger probability that a difference in QDC of only a few nT will move the range $r$ into a different bin. Because of this, it is also possible that the deviations are a result of the over and under-correction owing to the TGO QDV being averages over



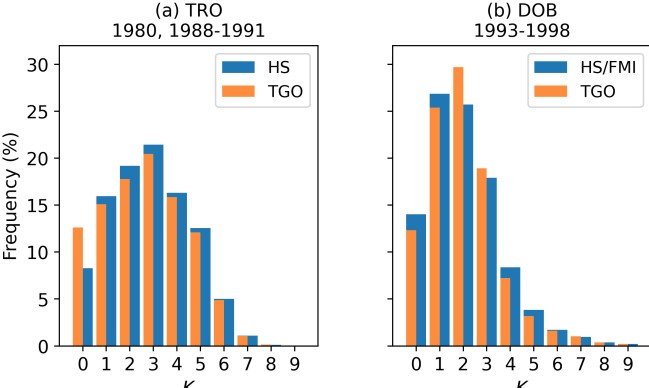

**Figure 4.** Distributions of $K$ during overlaps in HS/FMI derived $K$ and TGO derived $K$ for a) TRO (19080, 1988-1991) and b) DOB (1993-1998).

an entire solar cycle with the accompanying 10 nT uncertainity, which is bigger than the bin-size. For both DOB and TRO, the presented distributions will be complemented by one-on-one comparisons of the derived $K$ indices by the overlapping methods in a later section.

### 3.2 Distributions during maximum and minimum solar activity

It is a well known fact that geomagnetic activity peaks during the declining phase of the solar cycle and during solar maximum, and that the activity is at its lowest during solar minimum. It is therefore of interest to investigate the distributions of $K$, especially for TRO and AND, but also for the other stations during solar minimum and maximum, to find to what extent the distributions in Fig. 3 are influenced by the solar activity during the included years. Figure 5 shows histograms of $K$ during a solar minimum (2008-2010) and during a solar maximum (2001-2003).

It is of course expected that the shape of the distributions for $K$ will change with the degree of solar activity. We expect a shift toward higher $K$ values during maximum and a shift toward lower $K$ values during solar minimum. This is exactly what we see in the Fig. 5 and Table A3 for BJN, LYR and NAL. The general shape of the distributions do not change significantly, but we see the expected shift. Even though the there are shifts in the distributions, the distributions for both low and high solar activity are well tapered. We note zero frequencies for $K = 9$ for all three stations, but this is not unreasonable for quiet years as the average frequency is about 0.01 % for the three stations.

We see the opposite in the distributions for TRO and AND. The frequencies during solar minimum of both $K = 9$ and $K = 8$ are zero, and the frequency of $K = 0$ is almost 40 % for both stations. This skewed shape is not unique to this specific minimum, but reoccurs during other minima for both AND and TRO in the TGO derived $K$ indices. The same skewed shape is also present during most minima in the series of HS derived $K$ indices for TRO, covering many solar cycles. It is therefore likely that the shape of the skewed distributions for TRO and AND as seen in Fig. 3 is a feature of this strong solar activity



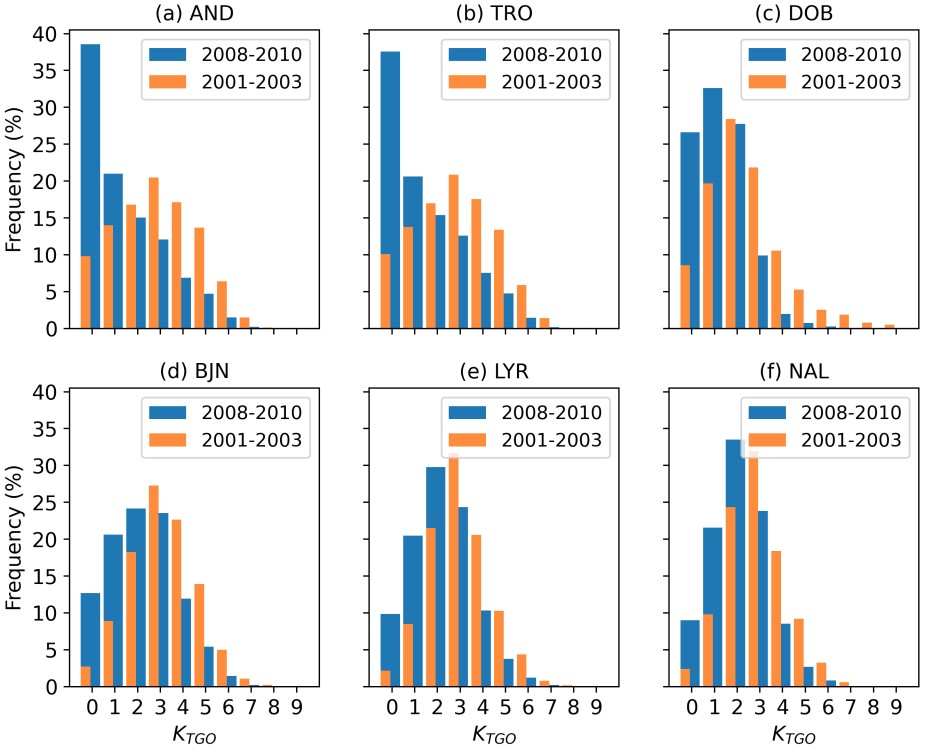

**Figure 5.** Distributions of TGO derived $K$ during a solar minima (2008-2010) and during a declining phase (2001-2003) for a) AND, b) TRO, c) DOB, d) BJN, e) LYR, f) NAL.

dependence, and not a result of some problem with the TGO method. This notion is strengthened also by considering the lower than usual levels of activity during solar cycle 24 (shown in Fig. 14).

The distribution for DOB, shown in Fig. 5c), shows the expected shift towards higher $K$ values during solar maximum. However, we see too high frequencies of $K = 9$ during solar maximum, and also during the solar minimum. These relatively high $K = 9$ frequencies in DOB might indicate that the $K9$-limit is set too low. A too low limit will yield a higher-than-expected

frequency of large $K$ values. The possibility that the K9-limit is too low was discussed as early as 1943 by Wasserfall during the first study of $K$ indices calculated for the DOB observatory (Wasserfall, 1943), and the discussion is clearly still relevant.

Similarly, the zero-frequencies of $K = 8, 9$ in AND and TRO might indicate that the $K9$-limit for AND and TRO is too high. This assumption is also strengthened by the fact that the $K9$-limit of 2000 nT is shared between AND, TRO and BJN. It is not unreasonable to assume that BJN experiences a higher level of geomagnetic disturbances. This is also evident from the

distributions for BJN shown in both Fig. 3 and Fig. 5. The limit of 2000 nT represents the variation at BJN well. Therefore it reasonable to assume that AND and TRO would benefit from a lower limit.



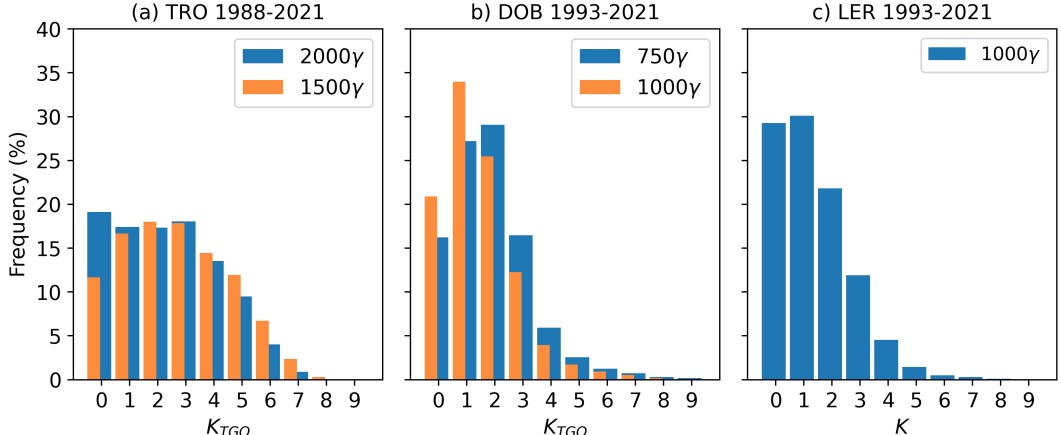

**Figure 6.** Histograms of K-values for a) DOB, K9-limit = 750 $\gamma$ (original) and K9-limit = 1000 $\gamma$ (adjusted),d) TRO, K9-limit = 2000 $\gamma$ (original) and TRO, K9-limit = 1500 $\gamma$ (adjusted) and c) LER, K9-limit = 1000 $\gamma$ (original).

### 3.3 Lower/higher limit for $K = 9$

Figure 6 shows the results of an experiment where distributions for DOB and TRO are recalculated using $K9$-limits of 1500 nT and 1000 nT, respectively. This was done by rescaling digitally stored magnetograms with the new lower limits, using the TGO

method. In Fig. 6a) it is clear that the distribution shape for TRO is very different, with peak frequencies at $K = 2$ and $K = 3$. The new limit of 1500 nT has not resulted in a too large frequency of $K = 9$, and the frequency of $K = 0$ is reduced. This clearly indicates that the new limit is a better fit for the variation experienced at TRO. The fact that the change in the distribution is most significant for $K$=0 means that this adjustment would not negatively affect the reasonable distribution shape for the entire series, shown in Fig. 3.

It is difficult to conclude whether the new limit for DOB is a better fit that the original limit. The frequency of $K = 9$ is slightly reduced, and the frequency of $K = 0$ increases. However, this increase in $K = 0$ is undesirable compared to the Niemegk distribution (Table 2). In addition, the frequency of $K = 9$ is still higher than we would expect. For comparison, Fig. 6c) shows the distribution of $K$ from 1993 to 2021 from Lerwick (LER), Shetland (retrieved from BGS Geomagnetism at http://www.geomag.bgs.ac.uk/data_service/data/magnetic_indices/k_indices.html). The K-9-limit for LER is 1000 $\gamma$, and LER

is only south from DOB by a few degrees in geomagnetic latitude. The distribution for LER is skewed toward low $K$-values. However, the frequency of $K = 9$ is similar to that for DOB with a $K9$-limit of 1000 nT. We cannot say that the K9-limit in DOB is too small, as the distribution does not necessarily improve with a higher limit. This is also indicated in Fig. 5, where it is clear that the distribution shape is robust to changes in solar activity, and therefore to changes in geomagnetic activity if we do not include the high $K9$-frequency.

It is shown that 1500 $\gamma$ limit is a better fit for the variation experienced at the TRO station. However, the limit cannot be replaced. For the sake of continuity from the manually derived $K$-values and preservation of the historic assignment of the




limit, the original $K9$-limit of 2000 $\gamma$ must be used. Nevertheless, the knowledge is valuable as it explains the unexpected distribution shape with the high density of $K = 0$ and the sensitivity to which years, and therefore which part of the solar cycle, is used when calculating the $K$-distribution. It also means that $K$-values from TRO and, say, DOB, does not necessarily

correspond to the same level of relative variation. In turn this means that the K-values cannot be used as a tool for directly comparing variation between the stations. Finally, an ill-fitting K-scale will not affect the validity of the K-series as measures of the time-variation of geomagnetic activity at each observatory. Therefore derived index ak and yearly Ak are still good measures of the time variation at each observatory, as will be shown later (Fig. 14).

## 4 Comparisons of $K$ derivation methods

During the overlap periods between *old* and *new* series for TRO and DOB, $K$ indices are derived by various methods. For DOB the overlap period (1993-1998) includes three methods, which includes overlaps between $K$-indices derived by the the TGO method and both HS and the FMI-method. For TRO the overlap period (1988-1998) also includes 3 methods, which are between $K$ indices derived by the the TGO method, an early version of the TGO method (the TGO' method) and hand scaling. A second overlap exist between HS derived $K$ indices and $K$ indices derived by the TGO method applied to digitized

magnetograms. Comparisons of the methods during overlapping periods are discussed in sections 4.1 and 4.2.

We have seen that the TGO method provides a good match with the HS and FMI derived $K$ values in the the *old* and *new* series for TRO and DOB (Fig. 4). However, it is also interesting to compare the performance of the TGO method against an acknowledged method. For this we chose the FMI method (Sucksdorff et al., 1991). The FMI method is acknowledged by IAGA (IAGA News, December 1993) and was found to be superior amongst the other methods considered (Menvielle et al.,

1995). The FMI method has been applied to the same data as the TGO method for this study. A comparison of resulting $K$ indices is presented in section 4.4.

### 4.1 Comparing $K$ derivation methods during overlap periods: DOB

For DOB, the TGO method and HS derivation overlap on the interval 1993-1994. The FMI method and the TGO method overlap during 1995-1998. Figure 7 shows the densities of all possible a) ($K_{HS}$,$K_{TGO}$)-pairs and b) ($K_{FMI}$,$K_{TGO}$)-pairs

during the overlap periods. For the $K$ indices derived by the FMI method, the yearbook for 1995 (Institute of Solid Earth Physics, Geomagnetism, 1997) notes that the $K$ indices for this year are primarily calculated by the FMI method, but that some indices are hand scaled. The same is noted in the yearbooks from 1996 (Institute of Solid Earth Physics, Geomagnetism, 1998), 1997 (Institute of Solid Earth Physics, Geomagnetism, 1999) and 1998 (Institute of Solid Earth Physics, Geomagnetism, 2000).

In general, the agreement between the methods in both comparisons are good, with clear abundances of perfect matches. However, about two thirds of the pairs deviate by one unit, and only 3 % of deviations by two or more units in both comparisons. In Fig. 7b) there is a large count of outliers. The most severe disagreement is seen in the upper left corner, where several $K_{TGO}$= 9 is paired with $K_{FMI}$={0, 1, 2}. However, all three points corresponds to the same day, that is the 25th of September 1998.





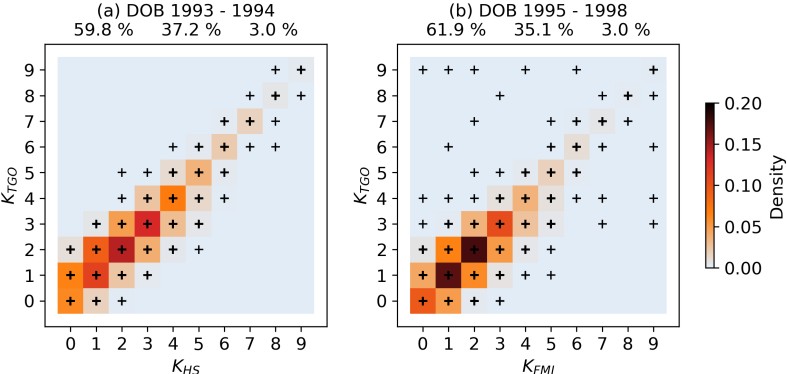

**Figure 7.** Comparison of a) HS derived $K$ indices and TGO derived $K$ indices and b) $K$ indices derived by the FMI method and TGO derived $K$ indices for DOB. The black crosses denote every pair with a density that is nonzero. Unmarked location have a density of zero. The frequencies of perfect matches, deviations by one unit and deviations by more than one unit are shown above each plot.

On this date the digital DOB magnetogram clearly exhibit stormy conditions. The same conditions are visible in the digital
TRO magnetogram. During this overlap period the DOB station operated two digital variometer systems (Institute of Solid
Earth Physics, Geomagnetism, 2000). The magnetograms that are stored digitally today at TGO, are results from the primary
variometer system. The published $K$ indices are calculated from magnetograms from the secondary variometer system. It is
therefore plausible that the $K_{FMI}$={0, 1, 2}, which are recorded in the DOB 1998 Yearbook (Institute of Solid Earth Physics,
Geomagnetism, 2000), are erroneous possibly due to a fault in the secondary variometer system or due to a human error.
The outliers where $K_{FMI}$=9 and $K_{TGO}$=4 in Fig. 7b), which occurred on the 4th of May 1998, can be explained by looking
at the stored, digital magnetogram from the same day. As explained in section 2.2.2, the TGO method will calculate $K$ based
on any interval containing data, no matter the size of the potential data gap. This can lead to erroneous, small $K$ values. This is
the case for the 4th of May 1998. The H component for this date is plotted in Figure 8. A large gap in the data has affected the
second (3-6 UT) and third (6-9 UT) intervals. In the second interval the variation is larger than the $K$9-limit of 750 nT, and the
TGO method therefore correctly assigns a $K$-value of 9 despite the missing data. In the third interval, the TGO method assigns
a $K$ value of 4 based on only the 25 minutes of data on the interval, even though the value could be larger based on size of
the data gap. However, the FMI derived $K$ value, based on the secondary variometer system which likely did not include this
data gap, assigns $K$=9. It is therefore likely that the $K_{TGO}$=4 is wrong and a result of the TGO method allowing any data gap.
This is also seen in other cases where the TGO method assigns a significantly lower value than both the FMI method and HS
derivation.
    The allowance of any size of data gap leads us to expect a tendency in which the TGO method assigns lower values for
$K$ than HS derived $K$. However, the opposite tendency can be seen in Fig. 7a), especially for low $K$ values. This is possibly
a result of the correction values being averaged over a solar cycle. This leads to over/under correction of the magnetograms,
especially during solar minima/maxima. Over correction will introduce false variation, whereas under correction will fail to



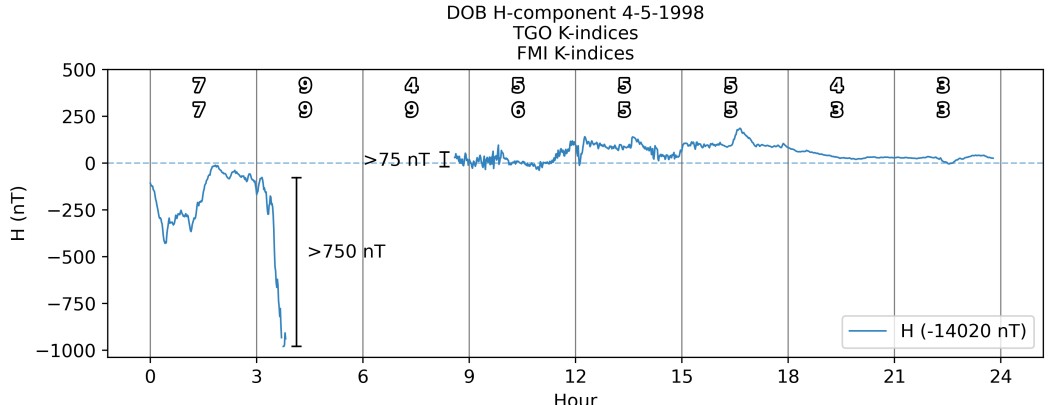

**Figure 8.** H component for DOB 4th of May 1998. The top row of indices are derived by the TGO method and the second row is deived by the FMI method.

remove all non-$K$ variation. Either way, the variation in the corrected magnetogram will be slightly larger than what should be expected. This has possibly resulted in the TGO method assigning larger $K$ values than the HS $K$ values, especially for the smaller $K$ values where the QDC have a larger influence on the range $r$ and therefore the assigned $K$. It is possible that it is necessary to implement less simple QDV that account for solar cycle variation of the $Sq$ current system and therefore the Quiet Day Variation. Another explanation for the shift toward larger $K_{TGO}$ for low $K$ values is an issue with the rigid QDV where

it is possible that shifts in time can lead to larger $K$ values. This is explored in section 4.4.3.

The deviations of one unit are symmetric between the FMI method and the TGO method (Fig. 7b). The yearbook from DOB 1995 (Institute of Solid Earth Physics, Geomagnetism, 1997) notes that the agreement between $K_{HS}$ and $K_{FMI}$ is generally good, with 75 % perfect matches. They also note that for the deviations of one unit $K_{FMI} > K_{HS}$ in 15 % of cases and that $K_{FMI} < K_{HS}$ in 10 % of cases. The comparison is performed for only a few months.

**4.2    Comparing $K$ derivation methods during overlap periods: TRO**

For TRO, the TGO method and handscaling overlap from 1988-1991. From 1992-1998 $K$ values based on the TGO and the TGO' methods are available. The difference between the two is that the TGO' method uses correction values derived from the most quiet days every month every year. In 1980 there is an overlap between hand scaled values and the values based on the TGO method applied to the same magnetograms, which exist in digitized form. Figure 9 shows the densities of all possible a)

$(K_{HS}, K_{TGO})$-pairs, b) $(K_{TGO'}, K_{TGO})$-pairs and c) $(K_{HS}, K_{TGO})$-pairs during the overlap periods.

From figure 9ab) it is clear that the agreement between $K$ indices derived by the two methods, during their overlap periods, is good. There is also a low number of outliers in all the comparisons, and the distributions are narrow, including less than 1 % of outliers of more than one unit. As is the case for DOB, the outliers for in $(K_{HS}, K_{TGO})$ can also be explained by (1) likely wrong $K$ value reported in the yearbook or (2) large data gaps leading the TGO method to give too low $K$ values.



All three distributions are asymmetric, showing clearly that for TRO the TGO method is prone to assigning lower values for $K$ than $K_{\mathrm{HS}}$, as is expected when data gaps are large in the TGO method. The signature of over and under correction that was visible in the DOB comparison in Fig. 7a), are not visible for TRO in Fig. 9a). It is likely that over and under correction still occurs, but we expect a significantly lower frequency due to the lower amplitudes of the QDC (Fig. 2) and the larger bins for $K$ which makes it less likely that under and over correction will affect the assignment of $K$.

In Fig. 9c), where the TGO method is applied to the digitized magnetograms from 1980, the asymmetry is stronger towards lower $K_{\mathrm{TGO}}$. This is possibly a feature of the digitization of the magnetograms. The digitization was performed by human hand and it is therefore likely that a bias is introduced.

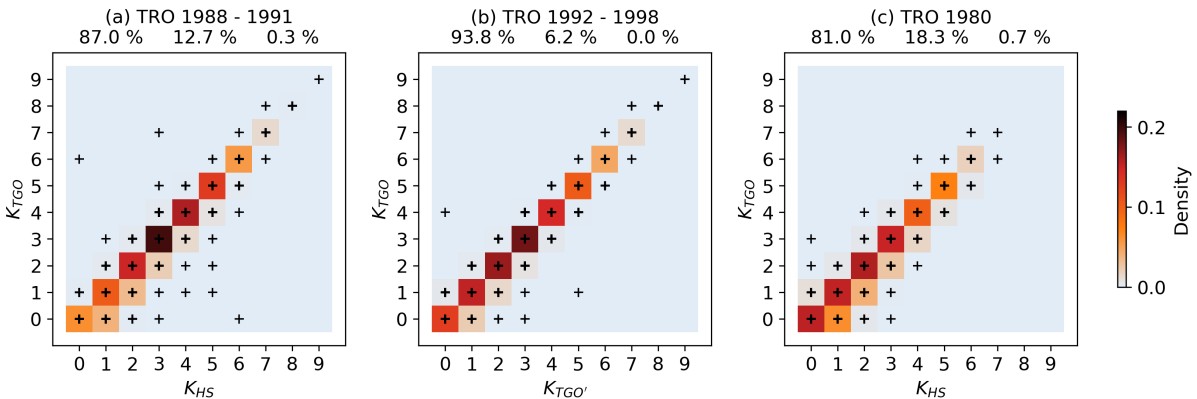

**Figure 9.** Comparison of a) HS derived $K$ indices and TGO derived $K$ indices, b) $K$ indices derived by the $TGO'$-method and TGO derived $K$ indices and c) HS derived $K$ indices and $K$ indices derived by applying the TGO method to digitized magnetograms for TRO. The black crosses denote every pair with a density that is nonzero. Unmarked location have a density of zero. The frequencies of perfect matches, deviations by one unit and deviations by more than one unit are shown above each plot.

The match between $K_{\mathrm{TGO}}$ and $K_{TGO'}$, shown in Fig. 9b) is almost perfect, with a percentage of 93.8 % of perfect matches and virtually no deviations above one unit. This is reasonable as the methods are identical expect the exact the correction

values. The asymmetry toward $K_{TGO'} > K_{TGO}$ indicates a systematic difference between $K_{TGO'}$ and $K_{TGO}$, likely due to the different correction values and under/over correction.

### 4.3   Summary of the compared methods

Generally, all five comparisons presented in Fig. 7 and Fig. 9 show good agreement. It is also evident that the agreements are better for TRO than for DOB. However, it is important to recall the $K$ distributions and the too high $K9$-limit for TRO. This

results in *less chances for errors* as a larger portion of the variation is *easily* placed in the $K = 0$ bin. As the bins for $K = 8$ and $K = 9$ are less frequently used, there are *smaller chances for errors*. We can see this in Fig. 9c). In 1980, neither $K = 8$ and $K = 9$ are used. This means that the variation is only between 8 bins.





It is also important to note that it is not clear whether these overlapping years are representative for all years. The overlaps are short and should ideally cover at least a solar cycle to be representative. It is therefore not possible to say if these comparisons produce results that are worse or better than for an average year.

In all the comparisons in Fig. 7a) and Fig. 9a) we see cases where $K_{HS} > K_{TGO}$ for all values of $K$. Apart from this tendency being a result of the TGO method assigning lower $K$ values to intervals with large data gaps, this is also likely a result of the transition from $K$ determination based on analogue magnetograms to applying the TGO method to 1 minute resolution data. Sampling in a digital system and computing 1 minute resolution data acts as a low pass filter. The magnitudes of distrubances in the digital 1 minute data will therefore be slightly smaller than the magnitudes of recordings on the analogue systems.

We see more $K_{HS} > K_{TGO}$ in DOB than in TRO. It is possible that this is a feature of the $X > Y$, or $H > D$ assumption that is applied in the TGO method. It is possible that the the assumptions is not valid for DOB since DOB is a sub-auroral station. In cases where the variation in $D > H$, the $K(D)$, which is calculated when hand scaling, will obviously be larger than $K(H)$, which is the only value calculated by the TGO method.

When evaluating $K$ derivation methods, IAGA accepted methods that achieved at least 69.9 % agreement with HS derived $K$ (Matzka et al., 2021) and less than 2 % deviations of more than one unit. Following these criteria, the TGO method is clearly an acceptable method when used at TRO. For DOB the two criteria are not met for the TGO method. However, the short overlaps and the fact that we cannot say whether the comparisons presented in this paper are representative for an average year, it is possible that the criteria could be met. It is also highly likely that adjusting the method, e.g. by including a max data gap length would increase the accuracy.

To summarize, we can confidently say that the TGO derived $K$ are a valid continuation of the HS derived $K$ for TRO. For DOB the same conclusion cannot confidently be made. However, as will be seen for both DOB and TRO (Fig. 14) that Yearly Ak is an excellent measure of the long term variation and that the agreement between the curves resulting from HS derived $K$ and TGO derived $K$ agree well.

### 4.4 The TGO and FMI methods

The FMI method has only been used in Norway during the short interval of 1995-1998 at DOB. For this study, the FMI method was applied to the digital TGO data. This was done, as mentioned, to compare the TGO method against a acknowledged $K$ derivation method, which the FMI method is. In this section we compare the FMI method to the TGO method, with and without applying the assumption $X > Y$. We also investigate a number of specific days and magnetograms.

Figure 10 shows comparisons of corresponding pairs of ($K_{FMI}$,$K_{TGO}$) for all 6 Norwegian $K$ index stations. The percentages above each plot are the percentages of perfect match between $K_{FMI}$ and $K_{TGO}$, deviations of 1 unit and deviations of >1 unit. Generally, it is clear that the agreement between the $K$ indices derived by the TGO method and the FMI method is good. For DOB the agreement is slightly worse than for the rest.

Most noticeable, all panels in Fig. 10 clearly show that the FMI method is prone to assigning large values for $K$ that does not match the TGO derived $K$ values, covering the entire range of $K_{TGO}$. The reasons for this was found to be due to a failure





of the FMI method when the QDC is fitted, leading to very large amplitudes. This can lead to erroneous, very large corrections and therefore ranges $r$, and finally also large $K$. Examples of these erroneous QDC are shown and discussed in section 4.4.2. These large $K_{\text{FMI}}$ contributes to the frequencies of deviations larger than one unit. However, we cannot solely blame the FMI method for this since we have already seen how the TGO method handles missing data leads to an assignment of lower values for $K$. It is, however, striking that the whole area below the diagonal is filled with mismatches, which leads us to conclude that the above mentioned problem with the FMI-method is the main reason.

The distributions for BJN, TRO and AND are clearly asymmetric, with a shift towards deviations one unit where $K_{\text{TGO}} > K_{\text{FMI}}$. It is not obvious why this is the case. However, Sucksdorff et al. (1991) notes that the FMI method is sensitive to disturbances during the night when estimating QDC. They show that the fitted QDC often follows the disturbances during the night, leading to a erronous QDC that also includes $K$ variation. For BJN, TRO and AND there are usually disturbances during the night, since they are all located in the auroral zone and the auroral electrojet is observed practically every night. These erronous QDCs can result in smaller $K$ values since $K$ variation is subtracted with the QDC. This can explain the strong asymmetry, and especially that we see the same asymmetry for both large and small $K$, which is different for the under and over correction in the TGO method which is only visible for lower $K$ values, as observed in Fig. 7a).

### 4.4.1 The X>Y assumption

To investigate the significance of the $X > Y$ assumption, the FMI method was also applied to TGO data with Y=X. Comparisons of $K_{\text{TGO}}$ and $K_{\text{FMI}}$ under the assumption that $X > Y$ are presented in Fig. 11, where some features are seen.

First, for BJN, AND, TRO, only small differences in the distributions and frequencies of the matches and deviations can be seen. This implies that the assumption $X > Y$ is valid at these locations. This is expected because these stations are in the auroral zone. For NAL and LYR there is a slightly higher increase in the percentage of perfect matches then when the assumption is not applied. We also note that the distributions are clearly asymmetric. The deviations toward higher $K_{\text{FMI}}$ have almost disappeared when comparing to the distributions shown in Fig. 10. This means that the these deviations were caused by $K$ indices based on $Y$ where $Y > X$. Further, this implies that for these cases, the $X > Y$ assumption is broken.

The same can be seen for DOB. The agreement improves drastically when the FMI method is applied with $Y = X$. As for NAL and LYR, the deviations of one unit where $K_{\text{FMI}} > K_{\text{TGO}}$ mostly disappears compared to the distribution shown in Fig. 10. Again, this implies that the deviations are the result of $K$ based on Y where $Y > X$ and that the assumption of $X > Y$ is broken.

### 4.4.2 Individual case comparisons between the TGO and FMI methods

Figure 12 shows three sets of three-day intervals of magnetometer X and Y components, $K$ values derived by both the FMI and the TGO method, as well as the range of the TGO method quiet curves and the QDCs fitted by the FMI method. The QDCs returned by the FMI method have been shifted by subtracting their means for easier comparison with the magnitudes of the TGO QDCs. Figure 12a) shows TRO data from center date 10-11-2022. This date is remarkably quiet. Under these conditions we see that both the FMI method and the TGO method assign only $K = 0$ and that their QDC are highly comparable. Figure



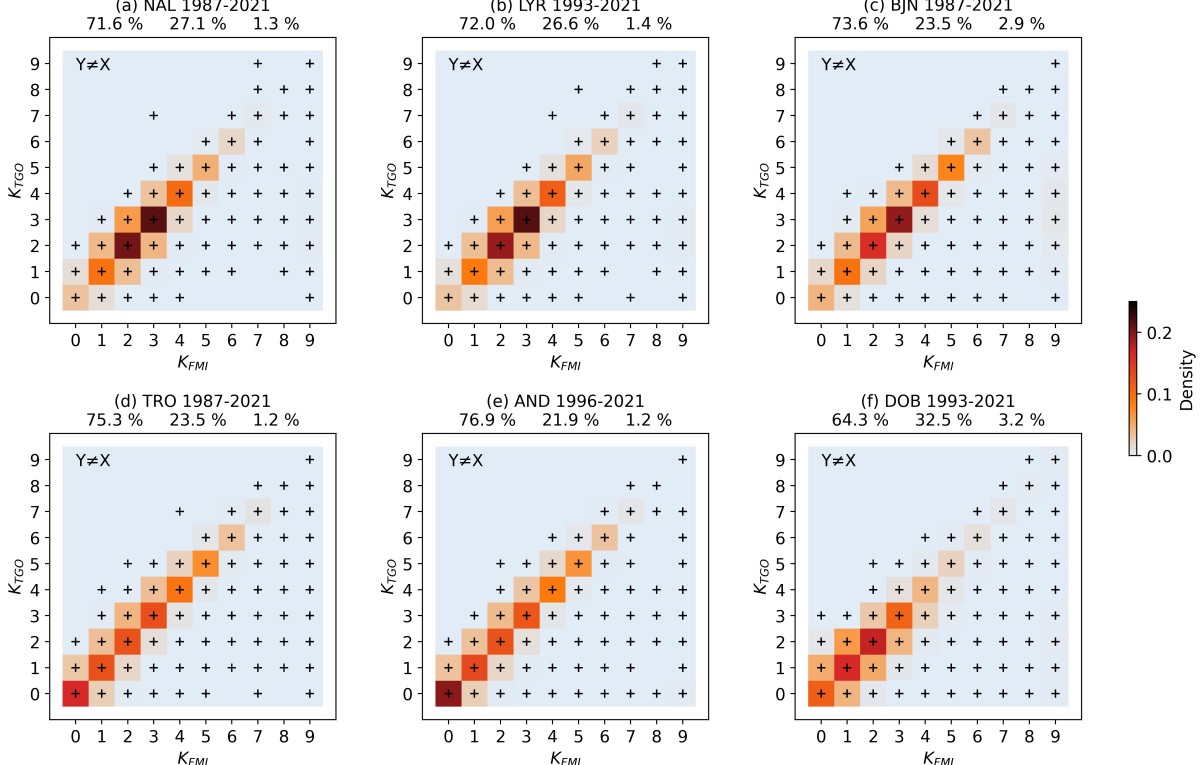

**Figure 10.** Comparison of $(K_{FMI}, K_{TGO})$-pairs for a) NAL, b) LYR, c) BJN, d) TRO, e) AND and f) DOB. The colors gives the density of each $(K_{FMI}, K_{TGO})$-pair occurring on the interval given in the title. The black crosses show every pair that have occurred. The percentages over each plot are the percentages of perfect match between $K_{FMI}$ and $K_{TGO}$, deviations of 1 unit and deviations of $>1$ unit.

12b) shows another quiet, but somewhat more disturbed, day for DOB (29-06-2021). For this case the $K$ values assigned by the FMI method and the TGO method are also identical and the QDCs practically comparable.

Next, Figure 12c) shows a magnetogram from TRO. For the three dates covered, 04-10-2022, 05-10-2022 and 06-10-2022, there are clear disturbances during nighttime. As mentioned above, a known limitation of the FMI method is the sensitivity to disturbances during the night when fitting the QDC. This is clearly visible in the Figure; the fitted QDC is clearly influences by

the disturbances in both shape and amplitude. For intervals 9-12 UT and 12-15 UT the assigned $K_{FMI}$ are smaller than $K_{TGO}$ by one unit, likely due to the too large QDC which has been subtracted. Disturbances during several successive nights is highly common at auroral latitudes, and this is not unique to the presented case.

There are a few cases where the FMI methods assigns a high number of $K = 9$, or $K$ values larger than $K_{TGO}$ by more than one unit (shown in Fig. 10 and Fig. 11). For these dates the fitted QDC are clearly erronous (not shown), with amplitudes up to

10000 nT. It is not clear why this error occurs, but for several cases that where investigated, there is at least one small gap in the data, in at least one of the three dates that are used for fitting the QDC. In the documentation of the C language version of





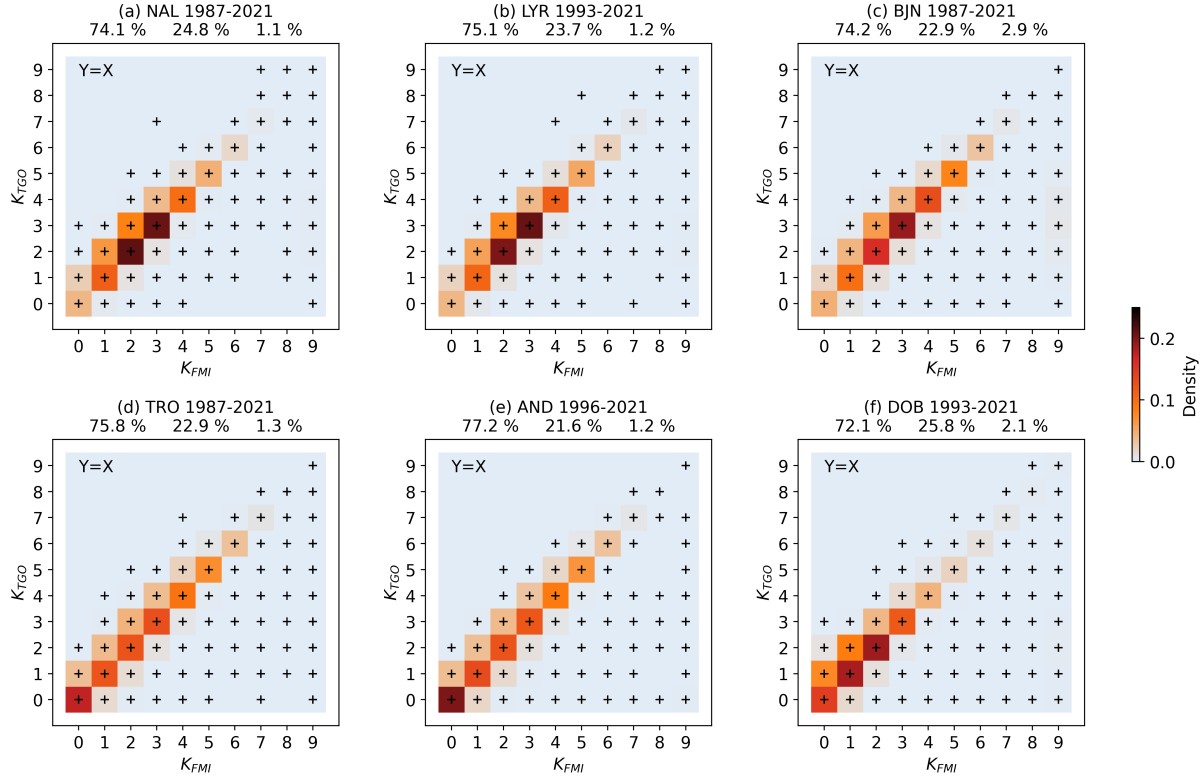

**Figure 11.** As Fig. 10. Here the assumption $X > Y$ is used and the FMI method is used with Y=X.

the FMI method, available on the FMI web page, they note that they have resolved a bug in version 1.1 of the program where erroneous results where returned if the previous day ended in missing data or when the center day started with missing data. The FORTRAN version available at ISGI is based on version 1.0, and it is therefore likely that the errors we see in our results

are also related to a known bug that is now resolved in the C language version, even though our data gaps occurs at other times than are noted in the documentation.

These kind of idetnified errors are rare. In the analysis done for this study every day of $K$ values for DOB and TRO where checked for cases where $K_{TGO} = \{0, 1\}$ where $K_{\text{FMI}} = 9\}$. Only 120 cases where found. The rarity of the error and the likely uncomplicated identification means that this error is not severe and does not reduce the reliability of the FMI method.

### 4.4.3   Differences in the $K$ variation

Figure 13 shows magnetogram H components, $QDC_{\text{FMI}}$ and $QDV_{\text{TGO}}$, and calculated $K$ variation for a) DOB 10-11-2022 and b) TRO 05-10-2022. These dates show two important cases when calculating the $K$ variation.

Figure 13a) shows a possible problem when dealing with rigid QDV. 10-11-2022 is clearly a quiet day for DOB when looking at the magnetogram. However, due to a phase shift in the quiet day variation on this date, there is a shift between




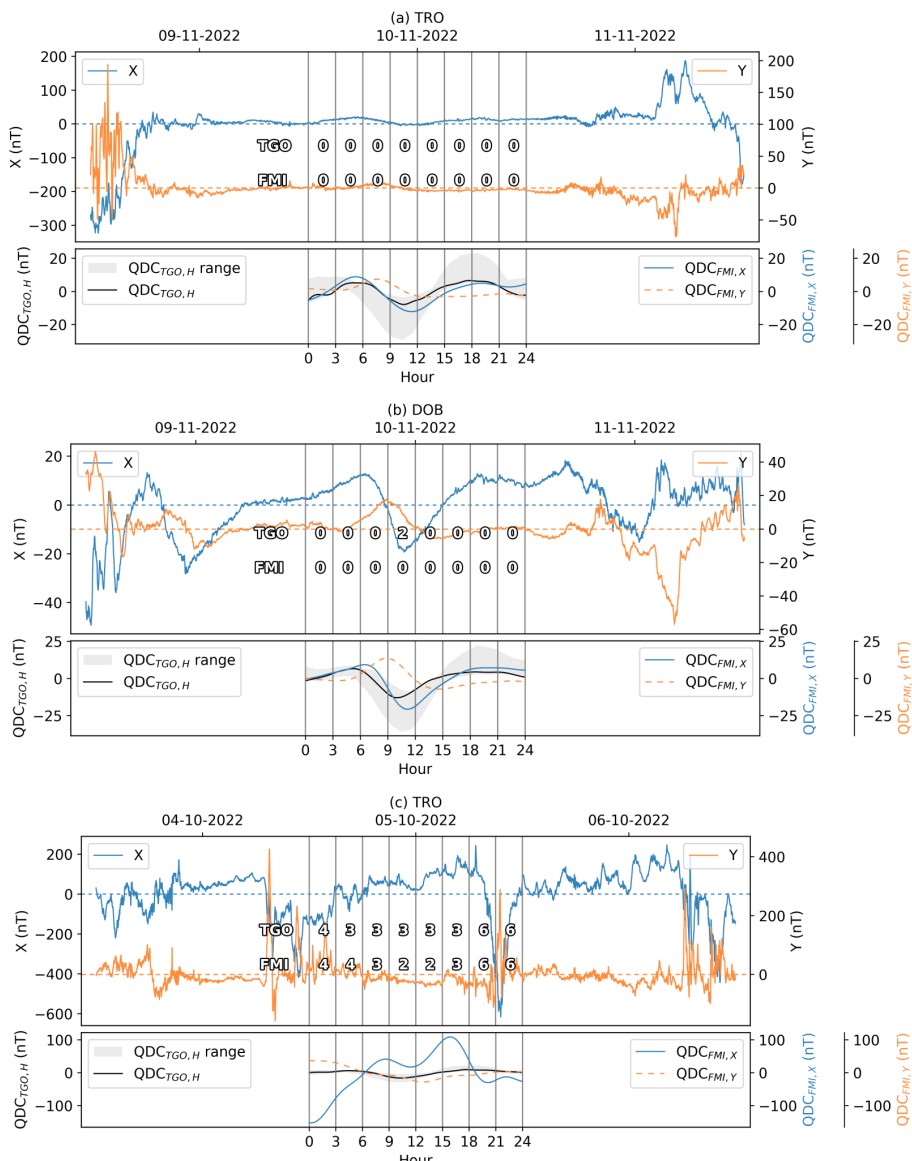

**Figure 12.** Magnetogram X and Y components, $K$ derived by the TGO and FMI methods, TGO method correction values and FMI QDCs for a) TRO 10-11-2022, b) DOB 29-06-2021 and c) TRO 05-10-22.

the H component and $QDV_{\mathrm{TGO}}$ leading to $K = 2$ in the 9-12 UT interval. This example is a "worst case", and shifts this severe, leading to an error of more than one unit, are rare. However, errors of one unit occurs regularly. This error is likely a contributing factor the asymmetry towards larger $K_{\mathrm{TGO}}$ for lower $K$ values in DOB, as seen clearly in Fig. 7a), and to why the agreement between the FMI and TGO methods is worse for DOB than for the remaining Norwegian stations. Shifts in the





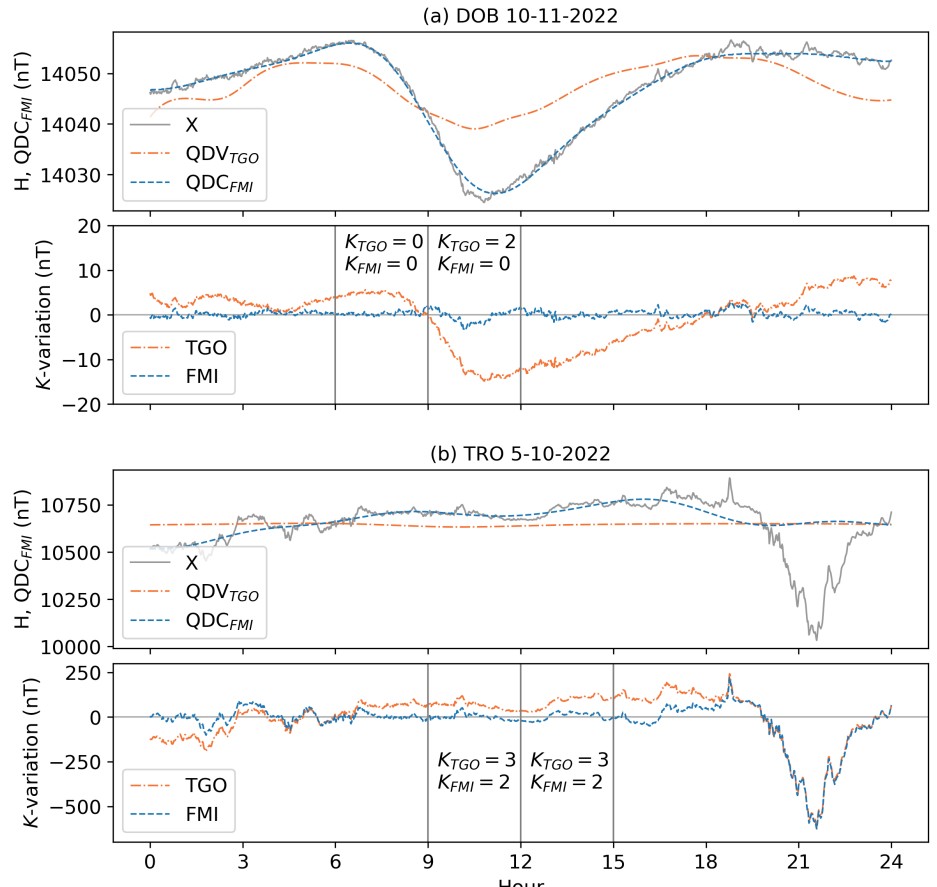

**Figure 13.** $K$-variation based on the QDC from the FMI method and the TGO method for TRO 10-11-2022 (top, as Fig. 12a), DOB-10-11-2022 (middle), and TRO 05-10-22 (bottom, as Fig. 12c)

quiet day variation in TRO also occurs. However, due to the larger bins for the low $K$ values for TRO, the assignment of $K$ is

rarely affected.

Figure 13c) shows the $K$ variation where $QDC_{\mathrm{FMI}}$ is affected by disturbances during the night. The magnetograms of the previous and next days are shown in Fig. 12b). Owing to the large magnitude of $QDC_{\mathrm{FMI}}$, the $K$ variation differs by up to 120 nT. For the intervals 9-12 UT and 12-15 UT the FMI method assigns $K = 2$, one unit lower than the TGO method. This effect is likely a contributing factor for the portion of deviations where $K_{\mathrm{TGO}} = K_{\mathrm{FMI}} + 1$ in Fig. 10.

**4.4.4   Summary of the comparison of the TGO and FMI methods**

We have seen that for all the Norwegian $K$ index stations except DOB the agreement between the FMI method is good and the expectation of at least 70 % correlation is met. For all stations except DOB and BJN the deviations above one unit are less





than 2 %. For BJN the percentage is slightly higher, but due to its extremely remote location this observatory is more prone to significant data gaps which likely have affected $K_{\mathrm{FMI}}$. For DOB, LYR, NAL we have seen that the assumption that $X > Y$
is broken often enough to yield visible differences in the comparisons when the FMI method is applied with an without the assumption.

Several causes for differences between $K_{\mathrm{TGO}}$ and $K_{\mathrm{FMI}}$ are summarized as follows. Disturbances during the night can result in large $QDC_{\mathrm{FMI}}$ and therefore $K_{\mathrm{FMI}} < K_{\mathrm{TGO}}$. Both shifts in the quiet day variation with respect to time and over and under correction can result in $K_{\mathrm{FMI}} < K_{\mathrm{TGO}}$. If the assumption $X > Y$ is broken and the variation in the Y component is larger,
$K_{\mathrm{FMI}} > K_{\mathrm{TGO}}$. Finally, we have seen that errors where $K_{\mathrm{FMI}} >> K_{\mathrm{TGO}}$ can occur, though seldom, when there are small data gaps in the data fed to the FMI method.

## 5 The $Ak$ time series

Figure 14 shows the complete series of the yearly averaged $Ak$ index for Bear Island (BJN), Dombås (DOB) and Tromsø (TRO). Note that for the curves to be more easily distinguishable, BJN and TRO are plotted with offsets of 15 and 7.5 nT,
respectively. The solid lines show the series obtained from the yearbooks and IAGA bulletins. For DOB and TRO these series include a transition period from HS derivation of $K$ to automatic derivation, as has been discussed earlier. The series for DOB includes a short interval where the FMI method was used, from 1995 to 1998. The dashed lines show the series resulting from the TGO method. The points for BJN and TRO in 1980 denote an $Ak$ resulting from applying the TGO method on digitized magnetograms that were digitized for a study by Walker et al. (1997). The yearly sunspot number (retrieved from WDC-SILSO
Sunspot Number, Royal Observatory of Belgium, Brussels https://www.sidc.be/silso/datafiles) is shown as grey shading.

The series from all three observatories generally express the same variation with time. All three series show expected local maxima during the declining phases of the solar cycle, and smaller local minima in the beginning of solar maxima. The yearly $Ak$ shows a clear global minimum in 2009, for all three stations, the most quiet year on record. The global maximum is reached in 2003 in TRO and BJN, but in 1991 for DOB. The global minimum in 2009 and global maximum in 2003 is shared with the
observatory in Sodankylä (SOD), Finland (Nevanlinna et al., 2011) which is closer to TRO than DOB in magnetic latitude.

The shape of the curve at BJN strongly deviates from the shape of the curves at TRO and DOB in 1958. The 1958 yearbook (Norwegian Institute for Cosmic Physics, 1960) makes no note of anomalies at the BJN observatory during the year 1958. Upon closer inspection of the monthly averaged $AK$ values for TRO and BJN it was revealed that there is a clear and systematic deviation between the two stations, only from January 1958 to June 1959. It is likely that this systematic deviation is of non-
physical origin. The scale values used at TRO are unchanged during this period, while the scale values for BJN are changed by only 0.1 nT (Norwegian Institute for Cosmic Physics, 1959, 1960, 1961). It is possible that the deviation is caused by human errors or that there were some differences in operations or routines at BJN during the IGY (1957-1958). We have not been able to identify any written references describing this issue.

The agreement of the overlapping series in TRO and DOB is generally good. The overlap in TRO is seemingly an almost
perfect match except a small difference in the first year, 1988. The difference between the overlapping series are larger in DOB,



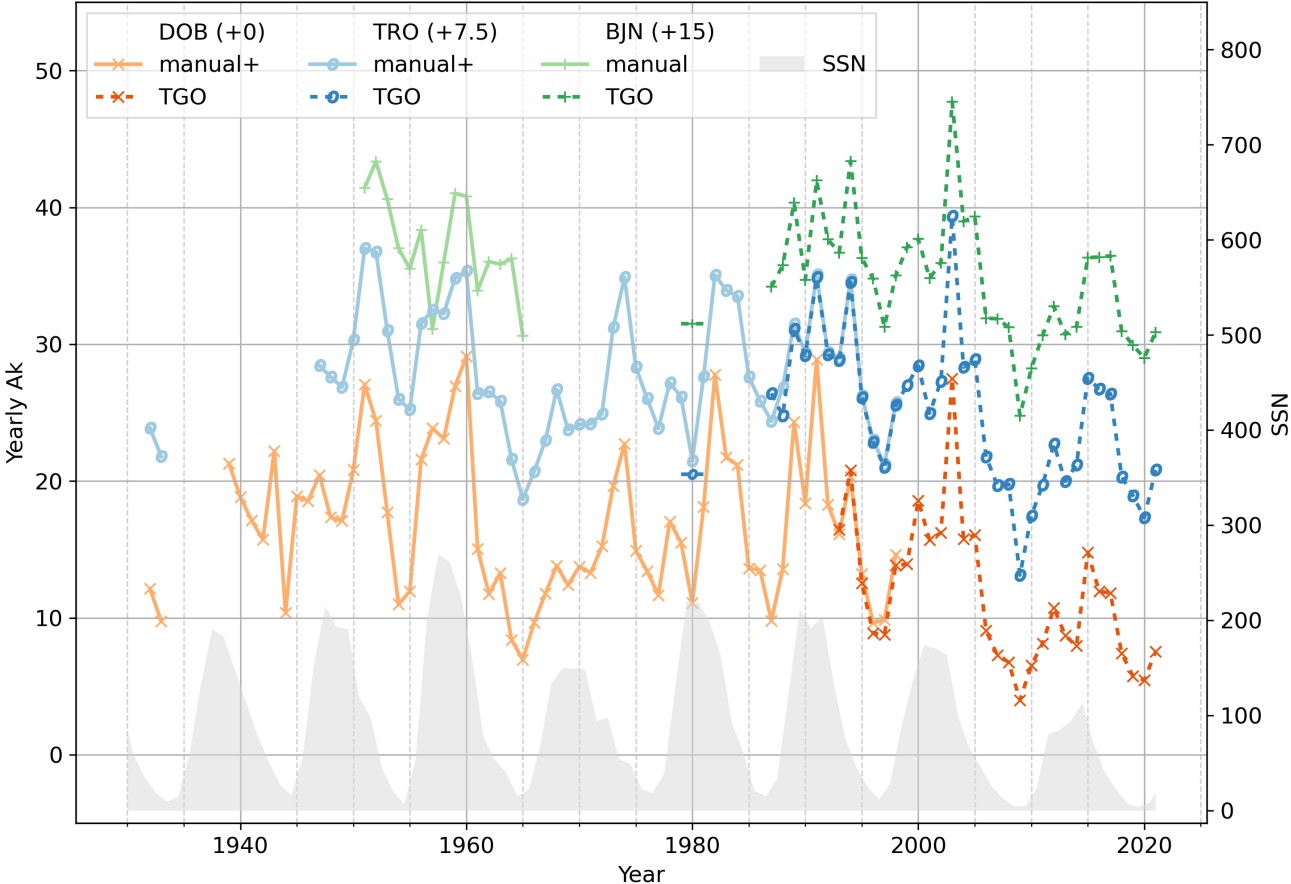

**Figure 14.** Yearly $Ak$ index from the stations in DOB (reds and oranges), in TRO (blues and purples) and on BJN (greens). The dashed lines show the automatically derived values. The solid lines show manually derived values. The "+" in manual+ denotes short transition periods from hand scaling to derivation by the TGO method.

with a maximum difference in 1997.

### 5.1 Power spectra of Daily $Ak$

Power spectra for Daily $Ak$ should exhibit several well-known spectral lines corresponding to solar and geomagnetic effects
(Matzka et al., 2021). Figures 15a) and b) show power spectra of Daily $Ak$ for TRO and DOB computed through Fast Fourier
Transforms (FFT) on the complete series. Both series only include a small number of gaps in the data. These gaps have been
filled with zeros. The complete series are constructed by simply joining the series of HS and FMI derived $Ak$ with the TGO
derived $Ak$. The series are joint in 1998 under the assumption that the series of TGO derived $K$ is a valid continuation of the


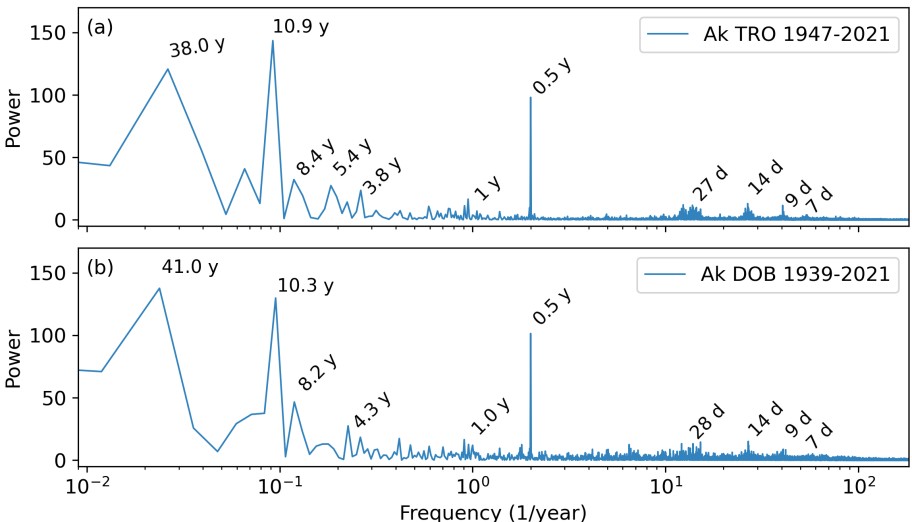

**Figure 15.** Power spectrum for the daily Ak series from (a) TRO 1947-2021 and (b) DOB 1939-2021.

series of HS and FMI derived $K$. As seen in Fig. 14, and which will be elaborated in a later section, this assumption is valid.
The power spectra for TRO and DOB are similar. The slight differences in positioning of the spectral lines can be accounted to the fact that the series for DOB is 8 years longer than the series in TRO, which corresponds to almost an entire solar cycle.

As expected, both spectra exhibit well-known spectral lines corresponding the the solar cycle, at 10.3 and 10.8 y, and the dual-peak structure of solar maxima, at 5.4 y, 4.3 y and 4.0 y. Both spectra also include strong lines at 8 y, at 8.4 and 8.2 y. It is possible that these lines correspond to the distance from the secondary peak of one solar cycle to the primary peak of the
next cycle. In addition, both spectra include strong spectral lines at 0.5 y corresponding to the biannual Russell-McPherron effect (Russell and McPherron, 1973). Both spectra also include spectral lines corresponding to the Earth's orbit, at 1.1 y and 1.0 y, and the solar rotation, at 27 d and 28 d. Furthermore, both spectra exhibit lines at harmonics of the solar rotation frequency, namely at 14 , 9 d, and at 6 . The peak frequency around each harmonic is rounded to the closest day to match the time resolution in our series, which is one day. The spectral lines at the first and second harmonic are believed to be caused
by coronal holes separated by 180 or 120 degrees in solar longitude (Temmer et al., 2007). The spectra for TRO and DOB include spectral lines around 6 d, which corresponds to the fourth harmonic of the solar rotation frequency. It is possible that these lines are due to four coronal holes separated by 90 degrees in solar longitude. Finally, both spectra show low frequency components for both TRO and DOB, at 37.6 y and 41.0 y, matching the low frequency components in similar spectra for Daily $Ak$ and Global Daily $aa$ (Nevanlinna et al., 2011).
The spectrum for DOB is clearly noisier than the spectrum for TRO, especially clear towards the higher frequency part of the spectrum. There are two possible explanations for this noisiness. First, DOB is located at a significantly lower geomagnetic latitude than TRO. It is therefore reasonable that solar-terrestrial interaction effects are more clearly visible in TRO than in



DOB. Second, it is likely that DOB data, especially up to the 1950s are influenced by various environmental factors that possibly influenced the recordings. These environmental factors are discussed in detail in e.g. Wasserfall (1953); Gjellestad et al. (1957). To check if these environmental factors have had a significant influence on the spectrum for DOB, the spectra were calculated on different intervals in time, excluding data up to 1952 and excluding all manual scaling. However, the noise level did not change noticeably.

## 6   Discussion and conclusions

In this paper we have reviewed the $K$ values generated for various locations in Norway during the last century and described where these have been published and how they have been calculated. We have also described the effort of digitizing all the identified K-indices. For the longest time-series at DOB and TRO, manual scaling of $K$ ended at about the same time as the transition from classic, analogue variometers to digital flux-gate magnetometers. Since these two observatories have been run by different institutes during most of their life times, and the documentation is limited, we have not been able to identify the practices of different scalers, we can only assume that they have followed the procedures as recommended in the literature (e.g. Mayaud (1980)). Several different approaches have been taken in the digital era for automatic determination of $K$ and limited periods of time contain overlaps between handscaling and automatic schemes. We have investigated these overlap periods in order to determine if the time series may be considered continuous accross the transition.

Investigating the frequency distributions of different $K$ values for the different stations, a few perculiarities were uncovered. In particular the auroral stations on the main land (TRO and AND) shows too many low K-values with the maximum at K=0 and too few high K-values, as compared to the "ideal" distribution (Table 4), i.e. the distributions are skewed towards low values. Furthermore, the frequency of K=9 is too high at Dombås. The skewness at TRO and AND was carefully investigated, where frequency distributions were examined for periods of high and low solar activity and for the possibility of scaling methodology introducing problems, but neither could be the reason. Comparing to neighbouring observatories, we realized that the K=9 threshold of 2000 nT chosen long time ago, could be the reason for the skewness, we made a test on the digitally available data and reduced the threshold to 1500 nT (same as for Sodankylä magnetic observatory). This improved the shape of the frequency distribution to satisfaction, leading us to conclude that the choise of threshold of 2000 nT is not justifiable. However, the threshold was set long time ago, when no statistical basis existed for its determination. Since the majority of data still only exist as analogue records and changing the threshold would require hand-scaling of more than 50 years worth of data, as well as the fact that the index has been used in previous studies, the threshold should be kept as is. This is no problem in itself, since it does not affect the time series analysis of the index itself, or its use in the analysis of other local time series. Furthermore, since the $K$ indices from TRO or AND are not combined with other stations to derive regional or global indices, no problems can be seen in that manner either. One should, though, show caution when comparing the $K$ from these stations with those from other stations, or at least be aware of the skewness in the distributions. Considering that already $K$ has a questionable value (e.g. Menvielle et al. (2011)) at high latitudes, since it does not necessarily reflect the magnetic energy density of the magnetic variations as it does at subauroral latitudes, we cannot problematize this further.



Scrutinizing frequency distribution for DOB made us suspect that the K=9 threshold of 750 nT is too low, especially considering that Lervick observarory situated somewhat further south, but nearby, uses a threshold of 1000 nT. Making the same test as for TRO and AND, recalculating $K$ with 1000 nT threshold did not improve the distribution, rather make it worse; although reducing the K=9 frequncy to an acceptable value, the maximum moved from K=2 to 1. Figure 6c includes the frequency

distribution for Lerwick, as can be seen the distribution is not "ideal" here either. It is possible that both Dombås and Lerwick would need an intermediate threshold between 750 and 1000 nT to satisfy the distribution provided in (Table 4). Again we point out that this issue with thresholds has little impact and do not affect treatment and analysis of the time series at a particlular observatory. Even so, in the case of Dombås, it is very useful to be aware of the unexpectedly high frequency of K=9, considering the wide usage of $K$ by the public looking for space weather proxies.

The comparison between hand scaled and automatic produced $K$ indices (TGO method) at TRO shows an agreement which is well inside the limits proposed by IAGA. This is the case both for a comparison of autoscaled indices based on digital data from a flux-gate magnetometer and handscaled magnetograms from a classic variometer system, and for a comparson between indices from autoscaling of a digitized, analogue magnetogram and handscaling of the same magnetogram from the classic variometer. Thus, we are confident that the $K$ index time series for TRO is homogeneous across the transition between the

analogue and digital eras. It is possible that the success of the TGO method for TRO relies on the relative large range bins for the different K-values, rather than the method itself being unusually elegant.

At DOB, however, the TGO method fails to comply with the recommendations of IAGA. Our investigations show that it is likely that this is because the TGO method assumes that X > Y, and at subauroral latitudes this is not necessarily a valid assumption, and therefore the method is less successful. Furthermore, the year of overlaps investigated here is during solar

maximum. Assuming that the automatic method would have less trouble with getting good matches with hand scaling during relatively quiet periods, it is plausible that the TGO method would also reveal better results if we had a greater range of years to compare. This notion is further strengthened by the fact that the good match using the FMI method reported in the 1995 yearbook (Institute of Solid Earth Physics, Geomagnetism, 1997), was only performed for a few months close to solar minimum, and redoing the scaling with the FMI method and comparing with hand scaling for the overlap period of Figure 7

reveals a perfect match of 69.6 % (not shown), which is just below the acceptance limit of IAGA. Thus, we conclude that if the overlap period between handscaling and autoscaling at DOB was longer, both methods would reveal better performance.

In our comparisons beteen the TGO method and the FMI method we identify different weaknesses and strengths. Using static and predefined Sq variation curves, as for the TGO method, does not allow for phase shift in Sq to be taken into account. We see that this introduces errors up to 2 units in $K$ during very quiet conditions at DOB. AT TRO the effect is less pronounced

owing to the larger ranges per $K$ unit and relative amplitude difference between Sq and other disturbances. Furthermore, the static Sq variation curves, does not take into account the fact that the amplitude of Sq in fact varies with the solar cycle, which will introduce a small inaccuracy that varies over approximately 11 years. Another weakness with the TGO method is how data gaps are handled. The method simply ignores them, which may cause a significant underestimation of the actual range. Considering that minimizing the amount of data gaps and continuous operation is one of the main objectives at magnetic





observatory, this problem is viewed as a minor one. Lastly, we find that the X > Y assumption usd by the TGO method is only
valid in the auroral zone, and does not hold at both at subauroral and very high latitudes.

The FMI method handles larger data gaps by avoiding them. However, small data gaps tend to create overflow in the Sq
estimate, bringing the resulting index far away from its real value. This might not be a problem with the newer version written
in C and available from FMI's web pages. Another issue with the FMI method we see in particular at high latitudes where the
magnetic field can be severely disturbed over many consecutive days, the method fails in the Sq estimate and produces too
small values.

In Figure 14, we finally show the complete digitized time series of Ak from DOB, BJN and TRO as yearly means. There is
generally, a good agreement between the three stations. We also see that the parts where there is an overlap between handscaling
and autoscaling, there is a good agreement between the curves. Of course, it is very likely that since the Ak is a mean, and
there is a close to symmetric scatter of autoscaled values around a handscaled "truth", the mean covering a whole year is fairly
robust against discrepancies between the two. In other words, although, there might be a break in the time series between
the two methods of scaling, in particular at Dombås, this break disappears when means are taken to present the whole time
series. The time series also reveal, as expected, the well known 2-3 year lag between geomagnetic activity and solar maxima,
which is explained by the presence of recurring high speed streams as well as an active Sun. Finally, the generated Ak curves,
corresponds well with those presented by Nevanlinna and Häkkinen (2010).

Reviewing the power spectra of the time series from Tromsø and Dombås, they are also in agreement, both with each other
and those presented by Nevanlinna and Häkkinen (2010) (AA, SOD Ak and Solar Wind velocity). It is seen that the spectrum
for Dombås is more noisy than that for Tromsø. We attribute this to a combination of Tromsø being in the auroral zone, and
thus seeing the effects of sun-earth interaction more frequently, and to the fact that the automatic scaling of K, is not optimal
at DOB.

We have seen in this work, that we have been able to bring forward long time series of geomagnetic activity from Norway.
We have demonstrated and documented the weaknesses and strengths of the time series in its current state. Little can be done
to correct for errors made during the hand scaling era without tremendous effort, however, relatively lightweight changes
and adjustments can be made to improve the time series in the future. The TGO method, has clear weaknesses, but it also
have a strength that it can be calculated in real-time. Small alternations to the method, such as better data gap handling and
abandoning the X>Y assumption, would improve the method. For operational space weather purposes, however, it is more than
good enough as it is, and it is very likely that it will be kept running as a provisional index. On the other hand, the FMI method,
which is acknowledged method by the scientific community, can easily be applied to digitally stored data, and it is therefore
desirable to establish a repository of K-indices generated using this method for the purpose of basic research purposes. This
would then be updated on a regular basis as data are quality checked and stored. The work reported in this paper, will serve
as a basis for the distinction between the two datasets. One should, however, be cautious about the fact that the Sq variation
determination, is difficult at auroral and polar cap latitudes owing to the presence of the auroral electrojet and polar current
signals eg. Lepidi et al. (2011) and Sillanpää et al. (2004) and through automated methods, in paritcular the Eastward electrojet,
often appear in the Sq curve, adding another uncertainty to the $K$ determination. One could possibly argue, that rather than





trying to elliminate the Sq current, owing to its inseperability with the Sqp current, and the relative size of other disturbances, one should just leave it in the data.



## Appendix A: K-index frequency distributions

**Table A1.** Frequencies of the distributions shown in Fig. 3.

| Station | Interval | Freq. $K$ [%] | | | | | | | | | |
|---------|----------|--------|--------|--------|--------|--------|--------|--------|--------|--------|--------|
| | | 0 | 1 | 2 | 3 | 4 | 5 | 6 | 7 | 8 | 9 |
| AND | 1996-2021 | 22.073 | 18.693 | 17.216 | 16.754 | 12.076 | 8.603 | 3.741 | 0.755 | 0.066 | 0.023 |
| TRO | 1932-1933 & 1947-2021 | 15.226 | 16.623 | 18.214 | 19.151 | 14.370 | 10.533 | 4.714 | 1.090 | 0.069 | 0.010 |
| | 1996-2021 | 21.590 | 18.243 | 17.434 | 17.361 | 12.538 | 8.533 | 3.532 | 0.703 | 0.053 | 0.013 |
| DOB | 1932-1933 & 1939-2021 | 16.633 | 22.257 | 25.862 | 18.977 | 8.394 | 3.989 | 1.971 | 1.121 | 0.504 | 0.293 |
| BJN | 1951-1965 & 1987-2021 | 4.503 | 12.096 | 21.466 | 27.576 | 19.104 | 10.789 | 3.793 | 0.613 | 0.051 | 0.007 |
| LYR | 1993-2021 | 4.503 | 13.829 | 26.940 | 29.507 | 15.542 | 6.552 | 2.471 | 0.551 | 0.090 | 0.014 |
| NAL | 1987-2021 | 4.502 | 14.564 | 28.847 | 29.769 | 14.084 | 5.785 | 1.966 | 0.431 | 0.042 | 0.010 |

**Table A2.** Frequencies of the distributions shown in Fig. 4.

| Station | Method | Freq. $K$ [%] | | | | | | | | | |
|---------|--------|--------|--------|--------|--------|--------|--------|--------|--------|--------|--------|
| | | 0 | 1 | 2 | 3 | 4 | 5 | 6 | 7 | 8 | 9 |
| TRO | HS | 8.278 | 15.953 | 19.192 | 21.452 | 16.309 | 12.571 | 5.026 | 1.096 | 0.110 | 0.014 |
| | TGO | 12.623 | 15.108 | 17.772 | 20.444 | 15.871 | 12.089 | 4.872 | 1.076 | 0.132 | 0.014 |
| DOB | HS/FMI | 14.024 | 26.847 | 25.721 | 17.925 | 8.380 | 3.849 | 1.702 | 0.958 | 0.369 | 0.225 |
| | TGO | 12.326 | 25.384 | 29.703 | 18.930 | 7.243 | 3.170 | 1.604 | 1.042 | 0.385 | 0.215 |



**Table A3.** Frequencies of the distributions shown in Fig. 5.

| Station | Interval | \multicolumn{10}{c}{Freq. $K$ [%]} | | | | | | | | | |
| | | 0 | 1 | 2 | 3 | 4 | 5 | 6 | 7 | 8 | 9 |
|---------|----------|-----|-----|-----|-----|-----|-----|-----|-----|-----|-----|
| AND | 2008-2010 | 38.570 | 21.008 | 15.064 | 12.052 | 6.893 | 4.713 | 1.501 | 0.199 | 0.000 | 0.000 |
| | 2001-2003 | 9.806 | 14.009 | 16.810 | 20.474 | 17.122 | 13.685 | 6.382 | 1.509 | 0.132 | 0.07 |
| TRO | 2008-2010 | 37.564 | 20.637 | 15.352 | 12.578 | 7.533 | 4.725 | 1.427 | 0.183 | 0.000 | 0.000 |
| | 2001-2003 | 10.084 | 13.758 | 16.985 | 20.865 | 17.523 | 13.380 | 5.872 | 1.373 | 0.080 | 0.080 |
| DOB | 2008-2010 | 26.631 | 32.630 | 27.737 | 9.888 | 1.973 | 0.741 | 0.274 | 0.091 | 0.011 | 0.023 |
| | 2001-2003 | 8.550 | 19.688 | 28.411 | 21.849 | 10.537 | 5.246 | 2.542 | 1.883 | 0.786 | 0.508 |
| BJN | 2008-2010 | 12.684 | 20.630 | 24.181 | 23.542 | 11.908 | 5.423 | 1.427 | 0.206 | 0.000 | 0.000 |
| | 2001-2003 | 2.738 | 8.881 | 18.246 | 27.264 | 22.643 | 13.898 | 4.992 | 1.065 | 0.223 | 0.050 |
| LYR | 2008-2010 | 9.838 | 20.492 | 29.791 | 24.359 | 10.338 | 3.749 | 1.184 | 0.210 | 0.039 | 0.000 |
| | 2001-2003 | 2.165 | 8.488 | 21.502 | 31.683 | 20.557 | 10.296 | 4.365 | 0.772 | 0.161 | 0.012 |
| NAL | 2008-2010 | 8.991 | 21.555 | 33.524 | 23.821 | 8.524 | 2.662 | 0.806 | 0.093 | 0.023 | 0.000 |
| | 2001-2003 | 2.393 | 9.811 | 24.327 | 31.963 | 18.397 | 9.181 | 3.217 | 0.607 | 0.080 | 0.023 |

**Table A4.** Frequencies of the distributions shown in Fig. 6.

| Station | K9-limit ($\gamma$) | \multicolumn{10}{c}{Freq. $K$ [%]} | | | | | | | | | |
| | | 0 | 1 | 2 | 3 | 4 | 5 | 6 | 7 | 8 | 9 |
|---------|---------------------|-----|-----|-----|-----|-----|-----|-----|-----|-----|-----|
| TRO | 2000 | 19.130 | 17.419 | 17.327 | 18.063 | 13.539 | 9.509 | 4.039 | 0.887 | 0.071 | 0.014 |
| | 1500 | 11.673 | 16.654 | 18.015 | 17.861 | 14.443 | 11.930 | 6.717 | 2.344 | 0.315 | 0.048 |
| DOB | 750 | 16.245 | 27.226 | 29.071 | 16.465 | 5.943 | 2.564 | 1.244 | 0.736 | 0.315 | 0.191 |
| | 1000 | 20.886 | 33.992 | 25.447 | 12.279 | 3.953 | 1.725 | 0.943 | 0.525 | 0.182 | 0.066 |
| LER | 1000 | 29.280 | 30.112 | 21.801 | 11.894 | 4.530 | 1.431 | 0.507 | 0.286 | 0.108 | 0.050 |



## Appendix B: Tables of $K$ index sources

**Table B1.** The sources of the digitized $K$ indices.

| Year | TRO | DOB | BJN |
|------|-----|-----|-----|
| 1939 | | Trumpy and Wasserfall (1944) | |
| 1940 | | Trumpy and Wasserfall (1944) | |
| 1941 | | Trumpy and Wasserfall (1944) | |
| 1942 | | Trumpy and Wasserfall (1949) | |
| 1943 | | Trumpy and Wasserfall (1949) | |
| 1944 | | Trumpy and Wasserfall (1949) | |
| 1945 | | Trumpy and Wasserfall (1949) | |
| 1946 | | Johnston et al. (1948a) | |
| 1947 | Howe et al. (1949) | Johnston et al. (1948b) | |
| 1948 | Howe et al. (1949) | Howe et al. (1949) | |
| 1949 | Bartels et al. (1950) | Bartels et al. (1950) | |
| 1950 | Bartels et al. (1951) | Bartels et al. (1951) | |
| 1951 | Norwegian Institute for Cosmic Physics (1953) Bartels et al. (1952) | Bartels et al. (1952) | Norwegian Institute for Cosmic Physics (1953) |
| 1952 | Norwegian Institute for Cosmic Physics (1954) Bartels et al. (1954a) | Bartels et al. (1954a) | Norwegian Institute for Cosmic Physics (1954) |
| 1953 | Norwegian Institute for Cosmic Physics (1955) Bartels et al. (1954b) | Bartels et al. (1954b) | Norwegian Institute for Cosmic Physics (1955) |
| 1954 | Norwegian Institute for Cosmic Physics (1956) Bartels et al. (1955) | Bartels et al. (1955) | Norwegian Institute for Cosmic Physics (1956) |
| 1955 | Norwegian Institute for Cosmic Physics (1957) Bartels et al. (1957) | Bartels et al. (1957) | Norwegian Institute for Cosmic Physics (1957) |
| 1956 | Norwegian Institute for Cosmic Physics (1958) Bartels et al. (1959) | Bartels et al. (1959) | Norwegian Institute for Cosmic Physics (1958) |
| 1957 | Norwegian Institute for Cosmic Physics (1959) Bartels et al. (1961) | Bartels et al. (1961) | Norwegian Institute for Cosmic Physics (1959) Bartels et al. (1961) |
| 1958 | Norwegian Institute for Cosmic Physics (1960) Bartels et al. (1962) | Bartels et al. (1962) | Norwegian Institute for Cosmic Physics (1960) Bartels et al. (1962) |
| 1959 | Norwegian Institute for Cosmic Physics (1961) Bartels et al. (1963a) | Bartels et al. (1963a) | Norwegian Institute for Cosmic Physics (1961) |
| 1960 | Norwegian Institute for Cosmic Physics (1962) Bartels et al. (1963b) | Bartels et al. (1963b) | Norwegian Institute for Cosmic Physics (1962) |



**Table B1.** The sources of the digitized $K$ indices (continued.)

| Year | TRO | DOB | BJN |
|------|-----|-----|-----|
| 1961 | Norwegian Institute for Cosmic Physics (1963) Bartels et al. (1964) | Bartels et al. (1964) | Norwegian Institute for Cosmic Physics (1963) |
| 1962 | Norwegian Institute for Cosmic Physics (1964) Bartels et al. (1965) | Bartels et al. (1965) | Norwegian Institute for Cosmic Physics (1964) |
| 1963 | Norwegian Institute for Cosmic Physics (1965) Bartels et al. (1966) | Bartels et al. (1966) | Norwegian Institute for Cosmic Physics (1965) |
| 1964 | Norwegian Institute for Cosmic Physics (1966) Prince et al. (1967) | Prince et al. (1967) | Norwegian Institute for Cosmic Physics (1966) |
| 1965 | Norwegian Institute for Cosmic Physics (1967) Veldkamp et al. (1968) | Veldkamp et al. (1968) | Norwegian Institute for Cosmic Physics (1967) |
| 1966 | Nordlysobservatoriet (1967) Veldkamp et al. (1969) | Veldkamp et al. (1969) | |
| 1967 | Nordlysobservatoriet (1968) Van Sabben et al. (1969) | Van Sabben et al. (1969) | |
| 1968 | Nordlysobservatoriet (1969) Van Sabben et al. (1970) | Van Sabben et al. (1970) | |
| 1969 | Nordlysobservatoriet (1970) Van Sabben et al. (1971) | Gjøen and Dalseide (1972) Van Sabben et al. (1971) | |
| 1970 | Nordlysobservatoriet (1971) | Gjøen and Dalseide (1973) | |
| 1971 | Nordlysobservatoriet (1972) | Gjøen and Dalseide (1974) | |
| 1972 | Nordlysobservatoriet (1973) | Gjøen and Dalseide (1975) | |
| 1973 | Nordlysobservatoriet (1974) | Gjøen and Dalseide (1975) | |
| 1974 | Nordlysobservatoriet (1975) | Gjøen and Dalseide (1976) | |
| 1975 | Nordlysobservatoriet (1976) | Gjøen and Dalseide (1977) | |
| 1976 | Nordlysobservatoriet (1977) | Gjøen and Dalseide (1978) | |
| 1977 | Nordlysobservatoriet (1978) | Gjøen and Dalseide (1979) | |
| 1978 | Nordlysobservatoriet (1979) | Gjøen and Dalseide (1980) | |
| 1979 | Nordlysobservatoriet (1980) | Gjøen and Dalseide (1981) | |
| 1980 | Nordlysobservatoriet (1981) | Gjøen and Dalseide (1982) | |
| 1981 | Nordlysobservatoriet (1982) | Gjøen and Dalseide (1983) | |
| 1982 | Nordlysobservatoriet (1983) | Gjøen and Dalseide (1984) | |
| 1983 | Nordlysobservatoriet (1984) | Gjøen and Dalseide (1985) | |
| 1984 | Nordlysobservatoriet (1985) | Gjøen and Dalseide (1986) | |
| 1985 | Nordlysobservatoriet (1986) | Gjøen and Dalseide (1987) | |



**Table B1.** The sources of the digitized $K$ indices (continued.)

| Year | TRO | DOB |
|------|-----|-----|
| 1986 | Nordlysobservatoriet (1987) | Gjøen and Dalseide (1988) |
| 1987 | Nordlysobservatoriet (1988) | Gjøen and Dalseide (1989a) |
| 1988 | Nordlysobservatoriet (1989) | Gjøen and Dalseide (1989b) |
| 1989 | Nordlysobservatoriet (1990) | Institute of Solid Earth Physics, Geomagnetism (1991a) |
| 1990 | Nordlysobservatoriet (1992) | Institute of Solid Earth Physics, Geomagnetism (1991b) |
| 1991 | Nordlysobservatoriet (1993) | Institute of Solid Earth Physics, Geomagnetism (1993a) |
| 1992 | Nordlysobservatoriet (1995) | Institute of Solid Earth Physics, Geomagnetism (1993b) |
| 1993 | Nordlysobservatoriet (1996) | Institute of Solid Earth Physics, Geomagnetism (1994) |
| 1994 | Nordlysobservatoriet (1997) | Institute of Solid Earth Physics, Geomagnetism (1995) |
| 1995 | Nordlysobservatoriet (1999a) | Institute of Solid Earth Physics, Geomagnetism (1997) |
| 1996 | Nordlysobservatoriet (1999b) | Institute of Solid Earth Physics, Geomagnetism (1998) |
| 1997 | Nordlysobservatoriet (1999c) | Institute of Solid Earth Physics, Geomagnetism (1999) |
| 1998 | Nordlysobservatoriet (2000) | Institute of Solid Earth Physics, Geomagnetism (2000) |



*Data availability.* The digital K-indices are available at the TGO website at https://flux.phys.uit.no/Kindice/Listindex.html. The digitized
K-indices are available at the TGO website at https://flux.phys.uit.no/Kindice/Manual/index.html. The sunspot number data is available
from WDC-SILSO, Royal Observatory of Belgium, Brussels at https://www.sidc.be/silso/datafiles. Lerwick data from BGS Geomagnetism
at http://www.geomag.bgs.ac.uk/data_service/data/magnetic_indices/k_indices.html. The FMI method FORTRAN code is available at The
International Service of Geomagnetic Indices (ISGI), $K$ indices software at https://isgi.unistra.fr/softwares.php.

*Author contributions.* IF performed the analysis, created all figures, digitized $K$ indices and wrote most of the initial manuscript. MGJ
initialized the project, supervised the work and wrote parts of the initial manuscript. Both IF and MGJ contributed to the interpretation of the
results, the discussion and the editing of the manuscript.

*Competing interests.* The authors declare that they have no conflict of interest.

*Acknowledgements.* We thank ISGI for the effort of scanning the IATME/IAGA bulletins and for making them and the FMI-method Fortran
program available online. We thank the British Geological Survey (BGS) for the Lerwick $K$ indices. We thank Alessandra Serrano and
Andrea Løkke for digitizing $K$ indices from the observatory yearbooks and IAGA bulletins. We acknowledge the work of the previous
and current observatory staff in Tromsø, Dombås and at Bear Island. Finally, we acknowledge the efforts of data processing and yearbook
preparation at the Auroral Observatory and at the University of Bergen.



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

Norwegian Institute for Cosmic Physics: The auroral observatory at Tromsø - observations 1951, Publikasjoner fra Det norske institutt for kosmisk fysikk, nr. 34, 1953.

Norwegian Institute for Cosmic Physics: The auroral observatory at Tromsø - observations 1952, Publikasjoner fra Det norske institutt for 850 kosmisk fysikk, nr. 36, 1954.

Norwegian Institute for Cosmic Physics: The auroral observatory at Tromsø - observations 1953, Publikasjoner fra Det norske institutt for kosmisk fysikk, nr. 37, 1955.

Norwegian Institute for Cosmic Physics: The auroral observatory at Tromsø - observations 1954, Publikasjoner fra Det norske institutt for kosmisk fysikk, nr. 38, 1956.

Norwegian Institute for Cosmic Physics: The auroral observatory at Tromsø - observations 1955, Publikasjoner fra Det norske institutt for kosmisk fysikk, nr. 40, 1957.

Norwegian Institute for Cosmic Physics: The auroral observatory at Tromsø - observations 1956, Publikasjoner fra Det norske institutt for kosmisk fysikk, nr. 41, 1958.

Norwegian Institute for Cosmic Physics: The auroral observatory at Tromsø - observations 1957, Publikasjoner fra Det norske institutt for 860 kosmisk fysikk, nr. 45, 1959.

Norwegian Institute for Cosmic Physics: The auroral observatory at Tromsø - observations 1958, Publikasjoner fra Det norske institutt for kosmisk fysikk, nr. 46, 1960.

Norwegian Institute for Cosmic Physics: The auroral observatory at Tromsø - observations 1959, Publikasjoner fra Det norske institutt for kosmisk fysikk, nr. 50, 1961.

Norwegian Institute for Cosmic Physics: The auroral observatory at Tromsø - observations 1960, Publikasjoner fra Det norske institutt for kosmisk fysikk, nr. 51, 1962.

Norwegian Institute for Cosmic Physics: The auroral observatory at Tromsø - observations 1961, Publikasjoner fra Det norske institutt for kosmisk fysikk, nr. 52, 1963.





Norwegian Institute for Cosmic Physics: The auroral observatory at Tromsø - observations 1962, Publikasjoner fra Det norske institutt for
kosmisk fysikk, nr. 53, 1964.

Norwegian Institute for Cosmic Physics: The auroral observatory at Tromsø - observations 1963, Publikasjoner fra Det norske institutt for
kosmisk fysikk, nr. 54, 1965.

Norwegian Institute for Cosmic Physics: The auroral observatory at Tromsø - magnetic observations 1964, Publikasjoner fra Det norske
institutt for kosmisk fysikk, nr. 55, 1966.

Norwegian Institute for Cosmic Physics: The auroral observatory at Tromsø - magnetic observations 1965, Publikasjoner fra Det norske
institutt for kosmisk fysikk, nr. 56, 1967.

Nozawa, S., Saito, N., Kawahara, T., Wada, S., Tsuda, T. T., Maeda, S., Takahashi, T., Fujiwara, H., Narayanan, V. L., Kawabata, T., and
Johnsen, M. G.: A statistical study of convective and dynamic instabilities in the polar upper mesosphere above Tromsø, Earth, Planets
and Space, 75, 22, https://doi.org/10.1186/s40623-023-01771-1, 2023.

Prince, A. T., Romana, A., Veldkamp, J., and IAGA: IAGA Bulletin No. 12s1 - Geomagnetic Data 1964, Indices K and C, IUGG Publications
Office, https://doi.org/10.25577/16nz-by06, 1967.

Russell, C. T. and McPherron, R. L.: Semiannual variation of geomagnetic activity, J. Geophys. Res., 78, 92–108,
https://doi.org/10.1029/JA078i001p00092, 1973.

Sergeyeva, N., Gvishiani, A., Soloviev, A., Zabarinskaya, L., Krylova, T., Nisilevich, M., and Krasnoperov, R.: Historical K index data
collection of Soviet magnetic observatories, 1957-1992, Earth System Science Data, 13, 1987–1999, https://doi.org/10.5194/essd-13-
1987-2021, 2021.

Sillanpää, I., Lühr, H., Viljanen, A., and Ritter, P.: Quiet-time magnetic variations at high latitude observatories, Earth Planet Sp, 56, 47–65,
https://doi.org/10.1186/BF03352490, 2004.

Sucksdorff, C., Pirjola, R., and Häkkinen, L.: Computer production of K-indices based on linear elimination, Geophys. Trans., 36, 1991.

Temmer, M., Vršnak, B., and Veronig, A.: Periodic Appearance of Coronal Holes and the Related Variation of Solar Wind Parameters, Sol.
Phys., 241, 371–383, https://doi.org/10.1007/s11207-007-0336-1, 2007.

Trumpy, B. and Wasserfall, K. F.: Results from the magnetic station at Dombås 1940 and 1941, Publikasjoner fra Det norske institutt for
kosmisk fysikk, nr. 23, 1944.

Trumpy, B. and Wasserfall, K. F.: Results from the magnetic station at Dombås 1942-1945, Publikasjoner fra Det norske institutt for kosmisk
fysikk, nr. 28, 1949.

Valach, F., Váczyová, M., and Revallo, M.: Producing K indices by the interactive method based on the traditional hand-scaling methodology
- preliminary results, Journal of Atmospheric and Solar-Terrestrial Physics, 137, 10–16, https://doi.org/10.1016/j.jastp.2015.11.009, 2016.

Van Sabben, D. and IAGA: IAGA Bulletin No. 32 - Geomagnetic Data 1970, Indices, Rapid Variations, Magnetic Storms, IUGG Publications
Office, https://doi.org/10.25577/fwxg-jq62, 1972.

Van Sabben, D., Siebert, M., and IAGA: IAGA Bulletin No. 12v1 - Geomagnetic Data 1967, Indices K and Ci, IUGG Publications Office,
https://doi.org/10.25577/tgtg-7h92, 1969.

Van Sabben, D., Siebert, M., and IAGA: IAGA Bulletin No. 12w1 - Geomagnetic Data 1968, Indices K and Ci, IUGG Publications Office,
https://doi.org/10.25577/t6bv-2a24, 1970.

Van Sabben, D., Siebert, M., and IAGA: IAGA Bulletin No. 12w1 - Geomagnetic Data 1969, Indices K and Ci, IUGG Publications Office,
https://doi.org/10.25577/kvcs-ey31, 1971.



Veldkamp, J., Siebert, M., and IAGA: IAGA Bulletin No. 12t1 - Geomagnetic Data 1965, Indices K and C, IUGG Publications Office, https://doi.org/10.25577/5qkt-zt70, 1968.

Veldkamp, J., Siebert, M., and IAGA: IAGA Bulletin No. 12u1 - Geomagnetic Data 1966, Indices K and C, IUGG Publications Office, https://doi.org/10.25577/4rcx-pq55, 1969.

Walker, J. K., Semenov, V. Y., and Hansen, T. L.: Synoptic models of high latitude magnetic activity and equivalent ionospheric and induced currents, Journal of Atmospheric and Solar-Terrestrial Physics, 59, 1435–1452, https://doi.org/10.1016/S1364-6826(96)00168-X, 1997.

Wasserfall, K. F.: Magnetic Horizontal Intensity at Oslo, 1843-1930, Terrestrial Magnetism and Atmospheric Electricity, 46, 173, https://doi.org/10.1029/TE046i002p00173, 1941.

Wasserfall, K. F.: Three-hour-range index, K, at Dombås observatory during 1939 to 1942, Terrestrial Magnetism and Atmospheric Electric-
ity, 48, 151–160, https://doi.org/10.1029/TE048i003p00151, 1943.

Wasserfall, K. F.: Results from the magnetic station at Dombås 1946, 1947, 1948, Publikasjoner fra Det norske institutt for kosmisk fysikk, nr. 35, 1953.

Zeller, O. and Bremer, J.: The influence of geomagnetic activity on mesospheric summer echoes in middle and polar latitudes, Ann. Geophys., 27, 831–837, https://doi.org/10.5194/angeo-27-831-2009, 2009.