# Peer review of "A critical review and presentation of the complete, historic series of K-indices as determined at Norwegian Magnetic Observatories since 1939"

_EGUsphere, 2024_

## Author Comment (AC1)

**Response to review on "A critical review and presentation of the complete, historic series of K-indices as determined at Norwegian Magnetic Observatories since 1939"**

*Ingeborg Frøystein and Magnar G. Johnsen*

**Response to reviewer 2**

**We thank the reviewer for reviewing our manuscript. Below we address their concerns, the comments are in black text and our responses are in red bold text.**

I find this work significant, since it offers not only a deep analysis of extensive time series of K, but also a thorough and detailed description from historical and analytical perspectives. Studies like this are valuable for dealing with series which result from concatenated measurements, something that is now widespread across numerous scientific fields dealing with long-term time series, like space weather and atmospheric and climate sciences, that have data which come from different instruments, analysts, or even changes in the measurment location.

I consider that this work can be accepted for publication in this journal with minor revisions. Below, I list some comments, followed by minor corrections I suggest regarding typographical or other minor issues.

Comments:

What is a "log-normal-esque" distribution? In particular, what does "esque" means?

**We agree that the "log-normal-esque" is an unfamiliar term, and we have changed it into "log-normal like" in the revised text.**

In section 3.1 (Distributions during the transition from handscaling to automatic methods) I suggest complementing the analysis including a statistical index with % confidence of the comparison made in Figure 4. Maybe the Kolmogorov-Smirnov test, to test the distributions similarities, or any other of your preference.

This will complement what you mention in Line 281. There you say " We have seen that the TGO method provides a good match with the HS and FMI derived K values in the the old and new series for TRO and DOB (Fig. 4)."

**We have performed a Pearson's test on our results, since it is robust with large data sets, and find high correlations (TRO 0.978 and DOB 0.893, respectively). We have added this to the revised manuscript.**

In Section 5.1 (Power spectra of Daily Ak), a possibility if the series has data gaps, is to use Lomb-Scargle. This is only a suggestion.

**We have made some rudimentary attempts on this, and found that no extra information can be extracted by this approach. We therefore prefer to avoid this extra effort. Also, for the sake of comparison with other studies like the one by Nevanlinna et al (2011), it is preferable to use the same technique. The data gaps in the time series'' are typically short and few (see https://flux.phys.uit.no/coverage/ if interested).**

In Figure 15, of the power spectrum, I would add the line showing the 95% confidence limit, or the red- or white-noise spectrum.

**We agree that a confidence line or noise limit could be useful. We add a line showing the red noise level and 95 percent confidence line to the spectra. In addition, we redo the spectra on log-log scale (instead of log-linear as in the originally submitted manuscript) to improve the readability of the spectra. We hare uncertain, however, if it adds much new information to the manuscript. Apart from showing that the labeled periods are real signals. We are interested in the reviewer's opinion on this.**

[Figure]

**Figure R1: New power spectra including a red noise distribution and a 95 percent confidence line.**

Minor comments:

Line 4: In "In t his paper," there seems to be a space in the worth "this". Please, check.

**Yes, that is correct. This is fixed in the revised manuscript.**

Lines 8-9: In " It becomes clear that each method "both have strengths and weaknesses." I am not sure if it should be "It becomes clear that each method have both strengths and weaknesses." Please check.

**We agree and have rewritten the sentence as "It becomes clear that each method has both strengths and weaknesses".**

Line 58: " Halddetoppen, marked in white." Isn't it in green?

**This has been fixed in the revised text.**

Line 79-80: " (e.g. (Nevanlinna, 2004; Nevanlinna et al., 2011)))" I think it should be "(e.g. Nevanlinna, 2004; 80 Nevanlinna et al., 2011)". Please check.

**This is now corrected.**

Line 116: " (e.g. Sergeyeva et al. (2021); Nevanlinna and Häkkinen (2010))." I think it should be " (e.g. Sergeyeva et al., 2021; Nevanlinna and Häkkinen, 2010)." Please check.

**This is now corrected.**

Line 151: " The QDVs are are the monthly averages ..." delete one "are". It should be " The QDVs are the monthly averages ..."

**This is now fixed.**

Figure 4 captions: "TRO (19080, 1988-1991)" should be "TRO (1980, 1988-1991)"

**This is now fixed.**

Line 227: "Even though the there are shifts in the distributions, ..." It should be "Even though there are shifts in the distributions, ...". That is, delete "the".

**Correct, this is fixed in the revised text.**

Figure 6 caption: Check this figure caption. a) corresponds to TRO and b) to DOB.

**The caption is now corrected.**

Line 434: "For these dates the fitted QDC are clearly erronous (not shown), with amplitudes up to 10000 nT.". Is 10000 nT the correct value? It seems too large. Please check.

**Yes, it is correct that the value was 10000 nT. This is why it was clearly erroneous. However, it has become clear that these QDCs are caused by a bug in the FMI fortran implementation. Following reviewer#1's suggestion, we have recalculated the FMI K-indices with the c language implementation instead. The results of our analysis are unchanged, but the cases where the QDC reaches values of 10000 nT and the FMI derived K-indices are too large are removed. Therefore, the discussion of these amplitudes is removed from the revised text.**

And "erronous" should be "erroneous".

**Yes, this is now corrected.**

Line 443: " where KFMI = 9}." Shouldn't it be " where KFMI = {9}." Please check.

**Correct, this is now fixed.**

Line 488: "AK" shoulfn't it be "Ak"?

**Yes, this is now corrected.**

Line 534-535: " (e.g. Mayaud (1980))." I think it should be "(e.g. Mayaud, 1980)." Please, check the instructions for authors.

**This is now corrected.**

Line 554: " (e.g. Menvielle et al. (2011))" I think it should be " (e.g. Menvielle et al., 2011)"

**This is now fixed.**

Line 559: "frequncy" should be "frequency"

**This is corrected.**

Line 561: "provided in (Table 4)." I think it should be "provided in Table 4."

**This is now corrected.**

Line 615: "have a strength" I think it should be "has a strength"

**Correct, this is now fixed.**

---

## Author Comment (AC2)

**Response to review on "A critical review and presentation of the complete, historic series of K-indices as determined at Norwegian Magnetic Observatories since 1939"**

*Ingeborg Frøystein and Magnar G. Johnsen*

**We thank the reviewer for reviewing our manuscript. Below we address their concerns, the comments are in black text and our responses are in red bold text.**

This manuscript studies geomagnetic K-indices measured in several observatories in Norway. The main author has digitized earlier indices from observatory yearbooks and IAGA Bulletins for tens of years, and calculated indices for the recent years using digital data. A detailed comparison is made for the early K-indices produced using hand-scaling and by two digital methods, the FMI K-index method and the local TGO K-index method (and its "preliminary" version). The manuscript studies the K-indices for several Norwegian observatories and discusses their long-term homogeneity. Authors have completed quite an extensive effort and, overall, the manuscript contains interesting information on the different K-index versions and methods to produce them. Therefore, the manuscript, eventually, may deserve to be published. However, I have several questions and even different interpretations on the results. Therefore, I have to suggest considerable modifications to the contents of the manuscript.

Presentation of the production of the digital K-indices using the two (actually, three) digital methods should form the solid basis of this work. Unfortunately, this basis is not laid very well and must be considerably improved. Authors say they used the Fortran code from IAGA to calculate the FMI K-indices. However, later, it becomes clear that this is an outdated code version which contains errors. Authors also elaborate later in the manuscript some problems due to those errors. The code has been corrected by FMI already a long time ago, implemented in a newer c code. It is not clear to me why the authors have used the outdated code version. If it is due to unfamiliarity with c language, this is no problem since codes can easily be transformed from one language to another by AI. I am confident that authors should indeed use the correct(ed) code. This will set all results on a more solid basis.

**The Fortran version of the FMI-method is the only version available from the International Service of Geomagnetic Indices (ISGI), which is the appropriate place to look for such software: https://isgi.unistra.fr/**

**This is also why we have used this version.**

**However, contemplating this, we have tested the C-version available from FMI, compared it with the fortran version, and found that they produce consistent results with very small differences (see left panel of the below figure which shows the difference between the two techniques). The only noticeable difference is the removal of the errenous K-indices where the FMI derived indices were much larger than the TGO derived indices (in Fig. 10 and 11 in the original manuscript). Since the C-version has been corrected for a bug, we have redone our analysis, and now present our results with basis in this version, in accordance with the reviewer's suggestion.**

[Figure]

**Figure R1: FMI-method: Difference between Fortran and C version (left), difference between using both X,Y and H,D (right). We have changed the text in the manuscript to reflect this change, and to briefly indicate that there are two versions available.**

Readers must be able to understand the different K-index methods in order to be able to evaluate the differences between the different K-index series. Now, the description of the FMI K-method is too short and unclear. E.g., it is not understandable how "These preliminary K indices are then used in calculation of the first fitted QDC" and how "These K indices are used in calculation of the final fitted QDC". This, and the whole recipe should be clarified, perhaps, using examples or figures in order to make the manuscript complete and understandable. It should be clear to the reader what sets the absolute level of the QDC curve and how its form will get determined. Also, I would like to have the problems related to the FMI QDC discussed here, not much later in the manuscript.

**We agree that it would be beneficial to include more careful descriptions of the K-index derivation methods. However, we will not include any new figures as the method is presented in more detail in papers by Suckcdorff 1991 and Menvielle 1995.**

**We point towards these two papers in the text for the interested reader and also to the FMI web pages where detailed examples are shown.**

**To improve the presentation of the three methods, we will in the revised text include one section per method (2.2.1, 2.2.2 and 2.2.3) and make sure that the procedure is more clear. In the revised text we make sure that the questions raised by the reviewer here are clearly answered.**

Even the description of the TGO method on pages 7-8 needs clarification. I dislike another concept of QDV, meaning the "values" used to calculate the TGO K-index, while they obviously are just hourly values of the (monthly) QDC. If this interpretation is wrong, please clarify. Else, simplify notation and remove QDV.

**We have removed QDV, and rather refer to the hourly average of the identified QDC throughout. This includes Figure 2, where we have updated the axis label to read QDC instead of QDV.**

I read that the H-component is used to calculate the QDC, so obviously TGO method uses this component for K-index calculation, although I did not find this clearly mentioned. On the other hand, page 7 says that the FMI method uses both X and Y-components to calculate respective K-indices (selecting the larger of them). The question is then, do authors compare K-indices that are calculated from different components in the two methods? As authors describe in the history part, during the hand-scaling time, both components were used (even Z originally), so the FMI method is more in line with long-term homogeneity. So why does the TGO method use the H-component, as it should be equally easy to continue the historical procedure? These issues are not much discussed in the manuscript.

**In the original definition of K, Bartels (1939) makes a point of not distinguishing between X,Y or H, D, This is furthermore repeated in e.g. Matzka et al (2021). After consideration, we agree with the reviewer that this distinction may be important, although aspects like the secular variation still may introduce problems anyway. We have therefore chosen to stick with H and D in all our K-value calculations, and made changes accordingly. Figure R1 in our answer above shows the agreement between XY vs DH in Tromsø.**

**For TRO, traditionally only H have been used, in the archives we have only been able to find an H gauge. However, as is presented in section 2.2.2 and investigated in section 4.4.1, the assumption of the disturbance in H always being larger than in D, is valid in the auroral zone.**

**The FMI method per definition, follows the offical procedure, and thus, uses both horizontal components (H,D or X,Y) in the K-index calculation.**

I would also like to see a more complete description of the TGO' method. (A better name would be, e.g. TGOp, p for preliminary).

**This have been done, see response above and section 2.2.3. TGO' is now referred to as TGOe (e for early).**

Another issue with notation is, e.g., X > Y and similar notations. I assume that this means that the K-index calculated from X-component is larger than the K-index calculated from the Y-component.

**This interpretation is correct. The X>Y notation refers to the assumption that the disturbance in the X-component is larger than in the Y component. In practice, this means that the K-index is calculated from only the X component, or now the H component (see above). In the revised manuscript, we will make sure that this is clear.**

This may belong to observatory jargon and be obvious for experts, but it must be clearly defined in the manuscript, as well as all other related notations. However, I still do not know for sure what authors mean by Y = X and Y ≠ X, which are used, e.g., in Figures 10-11. And what is meant in Sec 4.4.1 by "FMI method was also applied to TGO data with Y=X"?

**In figures 10 and 11, Y=X and Y≠X refers to the assumption that X>Y which is described in the previous answer. Y=X means that we assume that X>Y and the K-indices are calculated with the FMI method on only X. Y≠X means that the K-indices are calculated with the FMI method on both Y and X. This is made clear in the revised manuscript with the new text labels "Only H" (analogous to "only X" instead of "Y=X") and "Both DH" (analogous to "Both XY" instead of "Y≠X"). Also, to avoid the confusing X=Y notation in e.g. "The FMI method was also applied to TGO data with Y=X" we made changes in the text.**

Authors study in Sec 4.4.1 the validity of the X > Y assumption. I wonder why this is needed at all. Because the main aim of this research is to ensure historical homogeneity, it should be clear that the TGO method also uses the same two components (X and Y) and selects the larger of the two K-indices, just like it was done earlier. So, I am not sure if this discussion is needed. Moreover, as it was noted above, the TGO method uses the H-

component. So, again, why to test the X>Y for the FMI method? This is all quite confusing and badly motivated.

**We include this investigation into the X>Y (or H>D) assumption because the TGO method only uses one component since X(H) is generally larger than Y(D) while the FMI method uses both components. In this investigation, we show that using the FMI method on only X/H or with both XY/DH does not change the results for TRO. See also, previous point.**
**Regarding the components used, see previous responses.**

**In the revised manuscript we will include a more careful description of the methods and include a description of what components are included.**

In Sec 2.2.2 the authors describe the QDC choice for the TGO method. They have calculated a monthly set of daily curves averaged over the full solar cycle 22. I read this so that they did not use the actual quiet days at all. Anyway, this leads to a monthly set of so-called iron curves, which are then applied to observations made during all levels of solar activity. This is very questionable, and its motivation by ".. removing the possible problem of subjectivity when selecting quiet days.." does not weigh much in comparison to the problems that follow from this choice. Authors estimate that the solar cycle variation of the QDC amplitude is about ±10nT. Looking at Fig. 2 we can see that in winter months the typical amplitude is not more than ±5nT. Thus, if the solar cycle variation of the QDC curve was taken into account, it could have an essential effect on the K-index at certain times of the cycle.

**Indeed, the applied curves may be referred to as iron curves, and as is discussed in the manuscript, this may introduce certain errors. However, the curves are based on identified monthly QDCs that have been published in the yearbooks, and yearly averages have been constructed from these. The following figure (fig R2) shows the used QDCs as function of year, and the solar cycle variation becomes evident. We can read directly from the figure, the uncertainty introduced by taking the average. Since this have been thoroughly discussed in the text, we don't want to introduce this figure to the manuscript.**

**It is possible that during winter months the amplitude could become 100% larger than what is estimated in the TGO method. However, the amplitudes would still be negligible compared to the K-scale values, possibly this could change the K from 0 to 1, but hardly for the larger values.**

[Figure]

**Figure R2: Sunspot number (left), QDC as a function of year and hour (middle) and yearly minimum and maximum (right).**

In fact this may be the reason for the difference in the TRO K-indices between the TGO and TGO' methods (the latter taking the cycle variation into account).

**We make note of this solar cycle effect and resulting over/under-correction in section 4.1. We will again make a remark of the over/under-correction being caused by this solar cycle effect in 4.2.**

**It is well know that successive days of disturbed conditions in the auroral zone may create challenges in identifying a QDC that reflects the proper Sq variation. This also becomes evident in automatic routines such as the FMI method (Sucksdorff et al , 1991, also cited in the manuscript.) Thus, any attempt to isolate the disturbance field here, will be more prone to mistake compared to at lower latitudes, regardless of the approach.**

I also note that taking into account the cycle variation of the QDC curve is less important, not even necessary at auroral latitudes (see, e.g., Martini et al., JGR, 2016), but is important at sub-auroral and lower latitudes (like DOB), where the disturbance level is lower. Also, the effect of cycle varying amplitude may be different at auroral and lower latitudes.

**We agree, this has also been thoroughly treated in the manuscript in the description of the TGO method and in the discussion section (line 600 in original manuscript).**

It is slightly disturbing to read that the earlier version of the TGO method (TGO') did actually take the solar cycle effect into account. I wonder what made the TGO to change their method for, in my evaluation, a less appropriate version? On the other hand, it seems that authors already realize the importance of the cycle variation effect since "It is possible that it is necessary to implement less simple QDV that account for solar cycle variation of the Sq current system..".

**Work is underway to also make FMI-method calculated K-indices available on the TGO webpages. The TGO method will not be changed, however, owing to the fact that there are many studies, at least for auroral zone stations, who have used K-values derived this way at this point. The potential "harm" done in this is non-existent, as discussed in the paper, since the issue only introduces an uncertainty of less than one unit and is within the acceptance limits defined by IAGA.**

**The reason for the transition between TGO' (now TGOe) and TGO method was the desire to, in a simple way, calculate K in real-time as well as avoiding identification of the quiet day curve.**

Authors use considerable effort in trying to understand and even modify the skewed distributions of the more recent years. Alas, such modification is not needed and would even be harmful. The different distributions are due to the very weak solar activity of the last 15-20 years, when the Earth has spent most of its time in the heliospheric current sheet (Mursula et al., JGR, 2022) and the level of geomagnetic activity and storminess has considerably reduced. This has also modified the K-index distribution toward lower values. Trying to twist it back to the "normal" distribution of the more active times would be like using a medieval torturing machine. Luckily the authors seem to understand the correct explanation themselves. However, the treatment should better reflect this explanation. Authors could also check this using data from other (than Norwegian) stations.

**It is our opinion that investigating the frequency distribution of the indices at particular stations reveals the nature and what to expect from values generated at that site.  It is absolutely worthwhile to know the nature of the variations of the index that we are responsible for, it will also allow others to make better use of it.**

**After this exercise we have already been able to report back to colleagues from neighboring countries why DOB has a tendency to report higher K-values than at their observatories.**

**The observed skewness towards low K values is unique for TRO and does not only apply to recent years, both DOB (subauroral) and BJN (auroral proper) has a better log-normal like distribution also during low activity years. The point the reviewer brings up with an unusual large amount of time spent in the heliospheric current sheet during the last few solar cycles, is rather the reason why we discovered this peculiarity with the TRO index, than the reason why we should expect the distributions to be skewed to such extent at any station we have indicies from. In fact, far back in time when the K=9 thresholds were chosen (soon 100 years ago), limited statistical basis existed to justify the choice, and there was an urgency to get the index from as many stations as**

**possible. Therefore, qualified guesses were used, rather than stringent scientific approaches (Aud Chambodout, personal communication), and it is therefore not very surprising that we find such anomalies among K stations.**

**What we present here is a rather sober analysis of frequency distributions that does not appear as expected and differ from those of the other stations. As is stated, the objective of trying a different threshold, is not with the aim of correcting or modifying the index as produced at the station, but rather investigate and understand the nature of it.**

**The conclusion of the analysis does state that we need to keep the thresholds as they are, since they were set so long ago. We believe that in section 3.3 all these aspects are properly addressed.**

The analysis of the K9-limits at DOB and TRO are rather futile. Because of TGO method is different in component treatment from hand-scaling, the K9-limit estimates should only be made using the FMI K-indices. Anyway, on page 13 authors should more clearly conclude that increasing the DOB K9-limit is not reasonable.

**We will include DOB in the revised manuscript to reflect that there are good reasons to keep the same threshold there as for TRO.**

**We have also now run the FMI method on all the available digital data and made adaptations to the manuscript and our conclusions remain the same.**

**As for using only FMI indices for investigation of the K9-limits, the derivation method does not change the result. The following figure shows the same as 6a) but with FMI K-indices. The result is the same.**

[Figure]

**Figure R3: Figure 6a) in the manuscript with the FMI method instead of the TGO method.**

Many of the comparisons, e.g., those on the skewness of distributions, are rather vague and qualitative. Authors should aim to be able to present and use more quantitative

estimates. E.g., they should calculate the values of the skewness for the different distributions and, perhaps, even test their statistical difference.

**Per Reviewer #2's comments we have calculated Pearson's correlation coefficients for the distributions in Fig. 4. Overall, our approach has been to compare our distributions with the distributions found for Niemegk by Matzka et al (2021), since these form the basis for the original development of the K index. Since we are investigating the different stations' nature with this as a baseline, we think that the qualitative differences are sufficient.**

In several occasions, the text refers to a later section where the topic at hand will be explained or studied in more detail. This is a very annoying structure for the reader. The authors should try to write more directly, one topic at one time, rather than spreading one topic into several sections.

**We have gone through the manuscript and found passages where we refer to things like "as will be seen later" and forward referencing to figures (in particular nr. 14), and removed these and adjusted text accordingly.**

As explained, geomagnetic activity indeed maximizes at equinoxes, a fact which is known for 160 years (Sabine, 1956). The Russell-McPherron effect (projection of solar equatorial HMF field onto the geomagnetic dipole axis) contributes to this semiannual (not biannual) variation but it is not the leading contributor. The authors may find paper papers by Cliver, Lockwood, Lyatsky, Mursula, Newell, Yoshida and many others which study the semiannual variation and their three mechanisms (two other being equinoctial (dominant) and axial mechanisms).

**The reviewer makes a very valid point, and we acknowledge a somewhat sloppy treatment of this. We have changed the text accordingly: "... maximize during spring and autumn, which is consistent with the well-known semi-annual geomagnetic variation (Lockwood et al., (2020) and references within).**

**We have changed the word "biannual"" to semiannual and changed the reference to the same as above (Lockwood et al (2020)).**

I think that the part on Ak time series should be dropped out completely. The treatment of Ak time series and power spectra is currently rather elementary and does not contain anything new. It should be considerably expanded and elaborated in a different publication. I find that, rather, this paper should focus on its main topic of K-indices with contents modified along the lines indicated above. It will be more solid without the added Ak annex.

**We prefer to keep it, partly owing to the other referee's comment. Also, as shown by both Matzka et al (2021) and Nevanlinna (2011), displaying spectra shows that the time series behaves as expected. By showing the spectra of the Norwegian time series and comparing to those already published, we confirm that there are no systematic or fundamental problems compared to both Sodankylä and the Kp index.**

**We added reference to Matzka et al (2021) and their periodogram.**

Overall, it is very likely that the TGO K-indices, in their present version, cannot be used to continue the hand-scaling period indices. This is true for all stations, especially for the lower-latitude DOB. Rather, the continuation should be made with the FMI method K-indices.

**We do not agree in the referee's statement, and believe that we have discussed this throughout the manustript in an appropriate manner. However, we have also included more discussion on the application of the FMI method, and revised both our text and figures to be more consistent and stringent. (See reply to referee comments above). We agree that DOB yields the least successful results using the TGO-method, we have therefore emphasized in the text, that caution needs to be applied and the the FMI method is the preferred in future time series analysis for, at least, this station.**

Therefore, I am happy to read in the discussion that ".. FMI method… is therefore desirable to establish a repository of K-indices generated using this method for the purpose of basic research purposes". On the other hand, I cannot agree with the other statement "For operational space weather purposes, however, it (TGO method) is more than good enough as it is…". Rather, I suggest that the TGO method (maybe the TGO') should be using solar cycle variable QDC curves.

**As seen in Figures 10 and 11, there is a good match between the FMI method and TGO method. The majority of all the values are within one unit of each other. We have discussed elaborately in the text, that DOB is the weakest match. The TGO method would have been accepted according to the IAGA benchmark for TRO, but not for DOB. (see comparison between hand scaled and TGO-method scaled indices).**

**The FMI method does not work for real-time purposes, because one need the next day. The TGO' (TGOe), would also be challenging to implement in real time, since a certain amount of manual quality check in would be necessary in order to idetify the QDC each month. The TGO-method, as mentioned, does fit with FMI within +/- one unit (if the assumption is that the FMI method provides a perfect result), which then is the**

**uncertainty in the estimate provided in real-time. Also considering our findings regarding K=9 thresholds, we have an indication that this will cause a much greater uncertainty, than the applied method itself.**

Smaller notes

There should be another Table summarizing the availability of the different types of K-indices, including the years of availability and years analyzed (if different), the method used, the component(s) used etc.

**We believe that there are enough tables in the manuscript. Section 2.2, include the years the different methods have been used.**

TGO' is a bad name for the "preliminary" method. Better to use, e.g., TGOp (p for preliminary)

**Yes, see the earlier comment about this. We have changed the name to TGOe (e for "early").**

I would favour the notation of K_X, K_Y (K_H, K_D) in order to make it clear which component is used to calculate the index.

**We believe that with the added description of the components used in the method descriptions this is clear enough. We prefer the notation K_FMI, K_TGO, K_HS etc., to distinguish between the methods. Also including components in the notation would be cluttered.**

Fig. 1 Station abbreviations would show better in other color than white, and they could then be thinner.

**This is a matter of taste. We wish to keep the text as is, in white/black outline so the text is easily readable also in grayscale and for readers with color vision deficiencies.**

Table 4. Are the given K9-limits for the X component? What are they for H and Y? Please include them in the Table.

**K9-limits are for the maximum range in either H and D or X and Y. There is no dedicated threshold per component. See also earlier answers regarding components.**

Table 2. Correct the K=2 lower level.

**The correct lower level in Tab. 2 for K=2 should be 10 nT. This is corrected.**

Fig. 9. I see no point in including panel c here.

**The point here is that the published K values are hand scaled from the magnetogram. While the data later was digitized, and the TGO method has been applied to the digitized data. The plot shows that TGO method at Tromsø produces a very good match to the hand scaled (subjective) indices. The panel supplements panel a and adds a significant amount to the statistics.**

Figs. 12 and 13 could perhaps be joined. Enlarge Fig. 12 width.

**The figures have been adjusted somewhat, also tidying up the H-X confusion (see above). We prefer to keep them separated.**

I would prefer to use the 3-hourly time intervals as 00-02, 03-05,...with 21-23 as the last one, instead of the confusing 21-24.

**We prefer the intervals written as they are to mark the start/end of each interval, with 24 marking the end of the day as is standard in the 24 hour clock.**

Text additions like "See, e.g., .." on line 70 should be placed within parentheses. There are several of them.

**Yes, this is fixed.**

The fact that the FMI K-index method is supported by IAGA has been repeated several times. Once (in Sec 2.21.) is enough.

**This is cleaned up in the text.**

Write consistently K-index, sub-auroral, hand-scaling, over-correction, under-correction,..

**This is cleaned up in the text.**

Line 58. Haldde is marked in green, not white.

**Yes. This is corrected in the revised text.**

Line 115. If the early K indices were not listed in the yearbook, how did they enter in the IAGA Bulletin? I would like some clarification.

**It was not practice within the Norwegian community to calculate K indices, and they were therefore not included in the yearbooks. When the effort to generate K-indices from as many observatories as possible was started, the groups in Norway started reporting these directly on dedicated forms to Bartels. It took some years before K was included into the yearbooks. We have added to the text that for these years the K indices were submitted directly to the Association of Terrestrial Magnetism and Electricity (IATME, now IAGA).**

Line 132. "digital TGO data" should be more clearly defined here.

**Digital TGO data refers to data from TGO's digital flux-gate magnetometers. This is now clarified in the text.**

Line 296. "Two thirds" cannot be true since about 60% are perfectly matched (Fig. 7).

**Correct, this is a typo and should be "one third". This is corrected in the revised text.**

Line 367. Correct: In DOB, $K\_HS < K\_TGO$, but in TRO, $K\_HS > K\_TGO$. This difference is likely due to the different effect of cycle variation on QDC amplitude at the different latitudes.

**We rephrased the statement on line XXX, to clarify the meaning. The statements now reads: "The diagonal where $K\_HS = K\_TGO + 1$ is slightly stronger for DOB than TRO. ". The point here is, rather than solar cycle related, why the "count"/number of cases of KHS >KTGO in Dombås is larger than in Tromsø,**

The discussion related to solar cycle is found further up in the paragraph.

Line 425. Date is in error. The event is the same in Fig. 13b.

**Yes, the correct date, 10-11-22, is added to the revised text.**